# Scaling laws for language encoding models in fMRI

**Richard J. Antonello**
Department of Computer Science
The University of Texas at Austin
rjantonello@utexas.edu

**Aditya R. Vaidya**
Department of Computer Science
The University of Texas at Austin
avaidya@utexas.edu

**Alexander G. Huth**
Departments of Computer Science and Neuroscience
The University of Texas at Austin
huth@cs.utexas.edu

## Abstract

Representations from transformer-based unidirectional language models are known to be effective at predicting brain responses to natural language. However, most studies comparing language models to brains have used GPT-2 or similarly sized language models. Here we tested whether larger open-source models such as those from the OPT and LLaMA families are better at predicting brain responses recorded using fMRI. Mirroring scaling results from other contexts, we found that brain prediction performance scales logarithmically with model size from 125M to 30B parameter models, with ∼15% increased encoding performance as measured by correlation with a held-out test set across 3 subjects. Similar logarithmic behavior was observed when scaling the size of the fMRI training set. We also characterized scaling for acoustic encoding models that use HuBERT, WavLM, and Whisper, and we found comparable improvements with model size. A noise ceiling analysis of these large, high-performance encoding models showed that performance is nearing the theoretical maximum for brain areas such as the precuneus and higher auditory cortex. These results suggest that increasing scale in both models and data will yield incredibly effective models of language processing in the brain, enabling better scientific understanding as well as applications such as decoding.

Large language models have come to dominate the field of AI due to incredible capabilities that range from reasoning [1] to code generation [2] to even predicting how a human brain would respond to language [3]. Rapid improvement in these abilities has largely been driven by *scale*: the most capable models today use nearly identical architectures to early transformer language models [4], but have orders of magnitude more parameters and larger training data [5]. Overall, model capabilities–often measured as zero-shot performance across a range of language tasks–tend to scale logarithmic with the number of model parameters [6, 7], suggesting that improvements will continue as model scale increases. Here we test whether these scaling "laws" hold for the task of modeling the human brain.

The human brain is the quintessential language processing system, but there is still much to learn about how it processes and represents language. One paradigm used for this purpose is the *encoding model*: given measured brain responses to natural language, construct a model that predicts those responses from the natural language stimulus [8–19]. If an encoding model is able to accurately predict brain responses across a range of new stimuli, then that model must use similar representations to the brain. High-performing encoding models can then be interpreted to gain insight into the brain's computations [20–22] or the function of different brain areas [11, 23–25]. The highest performance is currently offered by encoding models that are based on large language models such as GPT-2 XL [26]. To build an encoding model, a language model is fed the same language stimuli that the

37th Conference on Neural Information Processing Systems (NeurIPS 2023).

human subject is hearing or reading. The internal states at each layer of the language model are then extracted, yielding *contextual embeddings* that capture semantic and syntactic properties of the stimuli [27]. These embeddings are then entered into a linear regression model that predicts the human brain responses, often measured using functional magnetic resonance imaging (fMRI).

Though text-based language models are the norm, language encoding models have increasingly been trained with acoustic features derived from audio-based neural networks [28–34]. Models like HuBERT [35] are able to derive phonetic, lexical, and semantic properties by learning from unlabeled waveforms or annotated transcripts [36]. Even when trained with human-plausible amounts of training data, these models can be more effective than language models in predicting brain responses in *low-level* speech processing areas such as the auditory cortex [31]. While earlier works examined the utility of several self-supervised audio models in brain encoding, newer models have since been released with substantially increased training data and speech recognition performance.

In this paper, we study whether encoding models for fMRI benefit from scaling neural network model parameters and datasets to the same degree as other tasks. We show that using contextual embeddings from larger language models can increase the prediction performance of encoding models by 15% over smaller counterparts. Larger acoustic models improve similarly with model size, showing largest improvements in auditory cortex and in higher-level areas. Finally, encoding performance for both model types scales logarithmically with the amount of fMRI training data from each subject, demonstrating an increasing need for very large fMRI datasets. These new state-of-the-art encoding models may enable a new frontier in the study of biological language comprehension and may provide deeper insight into the mechanisms that the brain uses to reason about and employ natural language.

## 2 Methods

### 2.1 Language models and speech audio models

Decoder-only transformer architectures have become dominant in recent years for language modeling [37]. For semantic encoding, we used representations from two families of large decoder-only Transformer language models, OPT [38] and LLaMA [39]. From the OPT family we used the pretrained 125 million, 1.3 billion, 13 billion, 30 billion, 66 billion, and 175 billion parameter models. From the LLaMA family, we used the pretrained 33 billion and 66 billion parameter models.

HuBERT and wav2vec 2.0 [35, 40] have been previously used to study auditory perception in the brain [29, 31, 33]. Both are trained to learn representations from unlabeled audio. WavLM [41] extends the HuBERT paradigm with data augmentation and also adds new data sources to increase the total training dataset size. Whisper [42] is a family of encoder-decoder models that use 680,000 hours of weakly-labeled audio – an order of magnitude larger than previous datasets – to reach state-of-the-art speech recognition performance. In this work, we used the pretrained Base, Large, and X-Large variants of HuBERT; the Base+ and Large variants of WavLM; and multilingual variants of the Tiny, Base, Small, Medium, and Large Whisper models.

Table 1 shows the architecture details for all neural network models used in this work.

### 2.2 MRI data

We used publicly available functional magnetic resonance imaging (fMRI) data collected from 3 human subjects as they listened to 20 hours of English language podcast stories over Sensimetrics S14 headphones [43, 44]. Stories came from podcasts such as *The Moth Radio Hour*, *Modern Love*, and *The Anthropocene Reviewed*. Each 10-15 minute story was played during a separate scan. Subjects were not asked to make any responses, but simply to listen attentively to the stories. For encoding model training, each subject listened to roughly 95 different stories, giving 20 hours of data across 20 scanning sessions, or a total of ∼33,000 datapoints for each voxel across the whole brain. For model testing, the subjects listened to two test stories 5 times each, and one test story 10 times, at a rate of 1 test story per session. These test responses were averaged across repetitions.

Details of the MRI methods can be found in the original publications [43, 44], but important points are summarized here. MRI data were collected on a 3T Siemens Skyra scanner at The University of Texas at Austin Biomedical Imaging Center using a 64-channel Siemens volume coil. Functional scans were collected using a gradient echo EPI sequence with repetition time (TR) = 2.00 s, echo time (TE) = 30.8 ms, flip angle = 71°, multi-band factor (simultaneous multi-slice) = 2, voxel size = 2.6mm x 2.6mm x 2.6mm (slice thickness = 2.6mm), matrix size = 84x84, and field of view =

Table 1: Model architecture summary.

| LANGUAGE MODELS | | | | AUDIO MODELS | | | |
|---|---|---|---|---|---|---|---|
| Family | Layers | Width | Parameters | Family | Layers | Width | Parameters |
| OPT [38] | 12 | 768 | 125M | Whisper [42] [a] | 4 | 384 | 8M |
| | 24 | 2048 | 1.3B | | 6 | 512 | 21M |
| | 40 | 5120 | 13B | | 12 | 768 | 88M |
| | 48 | 7168 | 30B | | 24 | 1024 | 307M |
| | 64 | 9216 | 66B | | 32 | 1280 | 637M |
| | 96 | 12288 | 175B | | | | |
| LLaMA [39] | 60 | 6656 | 33B | HuBERT [35] | 12 | 768 | 95M |
| | 80 | 8192 | 66B | | 24 | 1024 | 317M |
| | | | | | 48 | 1280 | 964M |
| | | | | WavLM [41] | 12 | 768 | 95M |
| | | | | | 24 | 1024 | 317M |

[a] Whisper counts include only the encoder.

220 mm. Anatomical data were collected using a T1-weighted multi-echo MP-RAGE sequence with voxel size = 1mm x 1mm x 1mm.

All subjects were healthy and had normal hearing. The experimental protocol used by [43, 44] was approved by the Institutional Review Board at The University of Texas at Austin. Written informed consent was obtained from all subjects.

In addition to motion correction and coregistration [43], low frequency voxel response drift was identified using a 2nd order Savitzky-Golay filter with a 120 second window and then subtracted from the signal. To avoid onset artifacts and poor detrending performance near each end of the scan, responses for training data were trimmed by removing 20 seconds (10 volumes) at the beginning and end of each scan, which removed the 10-second silent period and the first and last 10 seconds of each story. Test responses were trimmed by an additional 80 seconds (40 volumes) to account for an fMRI artifact (see Section 3.5). The mean response for each voxel was subtracted and the remaining response was scaled to have unit variance.

## 2.3 Encoding model construction

We used the fMRI data to estimate voxelwise brain encoding models for natural language using the intermediate hidden states of the various language and speech models discussed in Section 2.1. First, activations for each word in the stimulus text were extracted from each layer of each LM. In order to temporally align word times with TR times, we applied Lanczos interpolation together with a finite impulse response model [43]. Previous hidden state extraction methods (e.g. [23]) involved extracting the hidden state at the last token of the final word from a fresh context of fixed length of $N$ tokens. This method requires $N$ forward passes through the model in order to compute the hidden state for a single word. As this is impractical for models over a certain size, we improved computational efficiency here using a dynamically-sized context window. For a given story, contexts were grown until they reached 512 tokens, then reset to a new context of 256 tokens. More formally, the hidden state for token $i$, $H(i)$ is defined as

$$H(i) = \begin{cases} \theta\left(X_{(0,i)}\right) & i \leq 512 \\ \theta\left(X_{\left(256\left\lfloor \frac{i}{256}\right\rfloor - 256, i\right)}\right) & i > 512 \end{cases}$$

where $X_{(j,k)}$ is the context of the tokenized story $X$ from the token at index $j$ to the token at index $k$ and $\theta$ is the function parameterized by the language model. This allowed hidden state extraction for most tokens to be completed with a single forward pass per token, rather than $N$ forward passes as in previous methods. Differing tokenization schemes for handling whitespace across language models presented a challenge for consistent evaluation and were handled on a case-by-case basis.

Unlike the analyzed language models, the audio models used are bi-directional, so we must use a fresh context to preserve the causality of the extracted features. We windowed the stimulus waveform with a sliding window of size $16\,\mathrm{s}$ and stride $100\,\mathrm{ms}$ before feeding it into model. At each layer, we

use the hidden state of the final token as the model's representation for the window. As Whisper follows an encoder-decoder architecture, we only use states from the encoder, since it operates only on the waveform. We then downsample the features as before with Lanczos interpolation.

Let $f(H(\mathcal{S}))$ indicate a linearized ridge regression model that uses a temporally transformed version of the language model hidden states $H(\mathcal{S})$ as predictors. The temporal transformation accounts for the lag in the hemodynamic response function [9, 45]. We use time delays of 2, 4, 6, and 8 seconds of the representation to generate this temporal transformation. For each subject $s$, voxel $v$, and language model layer $h_i$, we fit a separate encoding model $f_{h_i}^{v,s}$ to predict the BOLD response $\hat{B}$ from our embedded stimulus, i.e. $\hat{B}_{(x,v,h_i)} = f_{h_i}^{v,s}(H_i(\mathcal{S}))$. Encoding model performance for a given layer was computed as the average voxelwise performance of that layer's hidden states across of all of cortex for all of our 3 subjects. For all figures with cortical flatmaps, we present the flatmap for one subject. Cortical flatmaps showing results for the other two subjects are shown in Section F of the supplement.

## 2.4 Stacked regression

A unified "optimized" encoding model combining the LLaMA language model and Whisper audio model was computed using an adaptation of the stacked regression approach from Lin et al. [46]. For every even-numbered non-embedding layer $l$ in the Whisper model, as well as the 18th layer of the 33 billion LLaMA model, we held-out $\sim 20\%$ of the training data and built an encoding model using the remaining $\sim 80\%$ of the training data. This was repeated for each of 5 folds. The predictions of these encoding models on the 5 folds of held-out training data were concatenated to generate full held-out predictions of the training data, $f_{h_l}^{v,s}\left(x_{h_l}^{(i)}\right)$. After this cross validation procedure, we build a covariance matrix for each voxel $v$ and subject $s$, $\boldsymbol{R}^{v,s}$ of the residuals such that

$$\boldsymbol{R}_{p,q}^{v,s} = \sum_{i=1}^{n} \left(y^{v,s} - f_{h_p}^{v,s}\left(\boldsymbol{x}_{h_p}^{(i)}\right)\right)\left(y^{v,s} - f_{h_q}^{v,s}\left(\boldsymbol{x}_{h_q}^{(i)}\right)\right)$$

where $n$ is the total number of time points and $y^{v,s}$ is the ground truth BOLD response for voxel $v$ on subject $s$. We then optimize the quadratic problem $\min_{\boldsymbol{\alpha}^{v,s}} \boldsymbol{\alpha}^{v,s\top} \boldsymbol{R}^v \boldsymbol{\alpha}^{v,s}$ such that $\alpha_{h_j}^{v,s} > 0$ and $\sum_{j=1}^{k} \alpha_{h_j}^{v,s} = 1$ with a quadratic program solver [47] to get a convex set of attributions $\boldsymbol{\alpha}^{v,s}$ which serve as weights for each feature space in the joint encoding model. This yields the final encoding model

$$\hat{y}^{v,s} = \sum_{j=1}^{k} \alpha_{h_j}^{v,s} f_{h_j}^{v,s}\left(\boldsymbol{x}_j\right)$$

where $k$ is the number of feature spaces. As a final step, we validate this stacked encoding model independently using a held-out validation set and build a final encoding model that uses the stacked prediction for voxels where the stacked approach is significantly better on this validation set and uses the prediction from the 18th layer of LLaMA otherwise.

To determine which layers of the model are used to model each voxel, we computed voxelwise attribution center-of-mass. For each of the $\boldsymbol{\alpha}^{v,s}$, the center-of-mass attribution $\mathcal{C}(\boldsymbol{\alpha}^{v,s})$ is computed as

$$\mathcal{C}(\boldsymbol{\alpha}^{v,s}) = \sum_{i=1}^{m} i\alpha_{h_i}^{v,s},$$

where $m$ is the total number of Whisper layers used in the stacked attribution. This allows us to summarize whether the attributions are primarily weighted on the earlier or later layers of the network for that voxel.

## 2.5 Noise ceiling computation

Data from brain recordings such as fMRI are inherently noisy, so it is useful to distinguish response variance that could potentially be explained by some model from noise variance that cannot be explained. We estimated the amount of explainable variance, or *noise ceiling*, using the averaged responses from the test story with 10 repeats and the method of Schoppe et al. [48]. The maximum correlation coefficient of the ideal encoding model is estimated as $CC_{max} = \left(\sqrt{1 + \frac{1}{N} \times \frac{NP}{SP}}\right)^{-1}$,

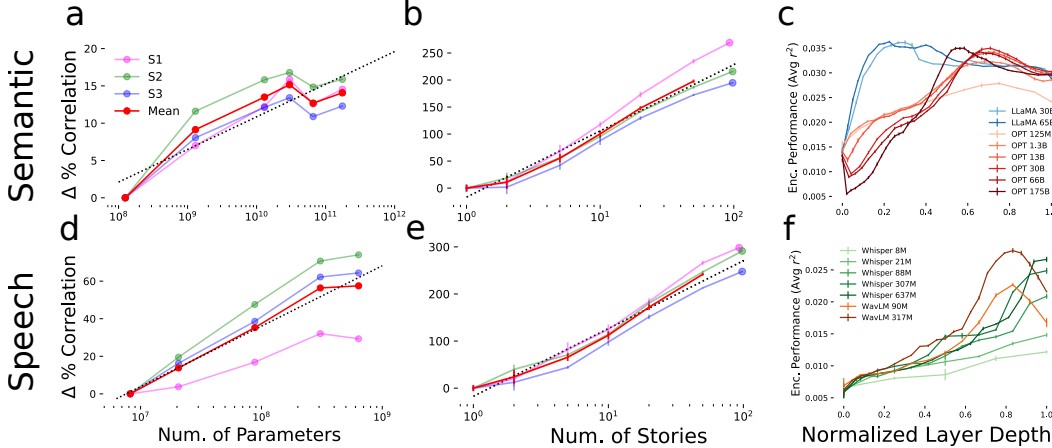

Figure 1: *Scaling laws of Semantic and Speech Audio Encoding Models* - **Figures 1a** and **1b** show logarithmic scaling of semantic encoding model performance with number of parameters and number of stories. **Figure 1c** shows average voxelwise $r^2$ for each layer of all tested models averaged across 3 subjects. **Figures 1d**, **1e**, and **1f** show analogous results for speech audio models. Error bars for **Figures 1b** and **1e** denote standard error across bootstraps. Error bars for **Figures 1c** and **1f** denote SNR-normalized subject-axis standard error. $r^2$ is computed as $|r| * r$.

where $N$ is the number of repeats of our test data, $NP$ is the noise power or unexplainable variance, and $SP$ is the signal power or the amount of variance that could in principle be explained by the ideal predictive model. Using these estimates, we can then extract a normalized correlation coefficient $CC_{norm} = \frac{CC_{abs}}{CC_{max}}$, where $CC_{abs}$ is the product-moment correlation coefficient of the model's predictions against the ground truth fMRI responses. In some voxels, random noise can cause $CC_{abs} > CC_{max}$, leading to $CC_{norm}$ estimates greater than one. To regularize $CC_{norm}$ estimates for noisy voxels we set $CC_{max}$ values smaller than 0.25 to 0.25. The normalized correlations $CC_{norm}$ are only used for the noise ceiling analysis in Figure 3. All other reported correlations are uncorrected ($CC_{abs}$). For brain map visualizations we only show voxels with $CC_{max} > 0.35$.

## 2.6 Compute specifications

The generation of the encoding models presented in this paper required significant computational resources. Ridge regression was performed using compute nodes with 128 cores (2 AMD EPYC 7763 64-core processors) and 256GB of RAM. In total, roughly 4,000 node-hours of compute was expended. Feature extraction from language and speech models was performed on specialized GPU nodes that were the same as the previously-described compute nodes but with 3 NVIDIA A100 40GB cards. Feature extraction required roughly 200 node-hours of compute on these GPU nodes.

## 3 Results

### 3.1 Scaling laws for semantic encoding models

Encoding models were fit for each of three subjects using roughly 20 hours of fMRI training data. For the 125 million parameter OPT model we also fit encoding models using varying amounts of training data in order to study the effect of training data size on encoding model performance. To capture encoding model performance, we compute the average prediction performance across all voxels in the cortex of each subject.

**Figure 1a** shows the relationship between language model size, measured as number of parameters, and encoding performance, measured as percent change in average prediction performance across all voxels in cortex relative to the smallest model. For consistent comparison, we only compare between the six model sizes from the OPT family. The layer that performed best for each model size was used. The result shows approximately logarithmic scaling of encoding performance with model size. For each order of magnitude increase in the number of parameters in the language, the encoding performance of the average subject increases by roughly 4.4%. However, this logarithmic relationship ($r = 0.91$) tapers off to a plateau for models in excess of ∼30 billion model parameters.

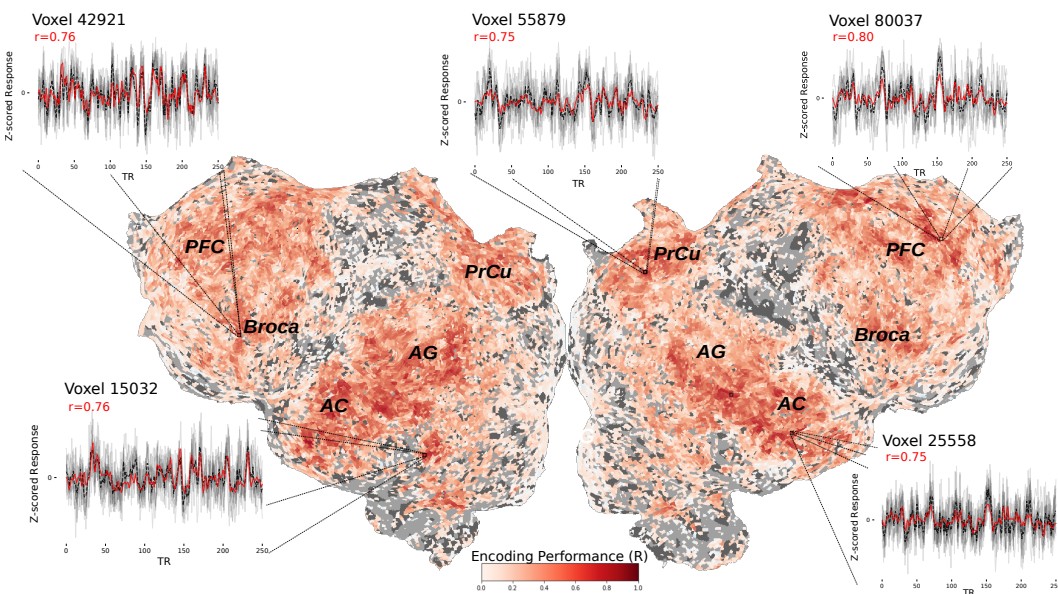

Figure 2: *Large-scale encoding models* - Performance of an encoding model built using OPT-30B on 20 hours of training data from a single subject. Surrounding plots show model predictions (*red*) against the average response (*dashed black*) over 10 separate trials (*gray*) on a held-out natural language test stimulus for selected voxels (*Clockwise from bottom left*: Well-predicted voxels from fusiform body area (FBA), Broca's area, precuneus, prefrontal cortex, and secondary auditory cortex.) Only voxels with $CC_{max} > 0.35$ are shown. (PFC = prefrontal cortex, PrCu = precuneus, AC = auditory cortex/Wernicke's area, AG = angular gyrus)

We hypothesize this is an effect of the increased hidden state size that larger models possess, combined with limitations in our dataset size. Each encoding model was fit using the same 33,000 data points. As the models grow wider, the regression problem becomes more ill-conditioned. When FIR delays are added, models past the 30B parameter threshold have more hidden states than there are data points to train the regression, which lowers encoding performance. Further analysis of the relationship between dataset size and model size is provided in Section E in the supplement.

**Figure 1b** shows the relationship between the number of training stories (roughly proportional to total training data) and encoding performance on OPT-125M (layer 9). Here we see a strong logarithmic relationship between training data size and encoding performance. Each time the number of training stories increases by an order of magnitude, the encoding performance of the average subject increases by 122%. This strong relationship ($r = 0.989$) gives compelling support to the usefulness of collecting "deep" datasets that focus on collecting a greater amount of data from a few subjects rather than a smaller amount of data from many subjects.

**Figure 1c** shows the encoding performance for each layer of each LM. The LLaMA models are marginally better at encoding than the OPT models, and also have a different pattern, with peak performance in relatively early layers followed by slow decay. In contrast, the OPT models have maximum performance with layers that are roughly 3/4 into the model. This mirrors results in other GPT-like models [3, 49]. This divergence from the typical pattern may warrant further study into the underlying mechanisms that define these trendlines. We suspect that the larger training set of the LLaMA models (1.4T tokens) over the OPT models (180B tokens) may explain its better encoding performance.

## 3.2 Scaling laws for speech audio encoding models

We trained encoding models of increasing sizes from three families of audio models: HuBERT, WavLM, and Whisper. Encoding models were fit using an identical procedure as with the LMs in Section 3.1 – individually for three subjects, with roughly 20 hours of training data. We repeat the analyses from Section 3.1 on the audio models to examine the importance of model size and training dataset size on encoding performance.

**Figure 1d** shows how audio model size affects encoding performance. We use the Whisper model family for this analysis, since it has the most models of different sizes. Again, the best performing layer for each size was used. As before, there is a logarithmic relationship ($r = 0.991$) between model size and encoding performance; performance for the average subject increases roughly 32.2% for every additional order of magnitude increase in model size. Though the scaling improvements are greater overall than with OPT, it should be noted that the smallest Whisper models are substantially smaller than the OPT models, and have lower baseline performance, which exaggerates the difference. Additionally, within auditory cortex, we observe that encoding performance does *not* plateau with model size (see Section B.1), suggesting that improvements in AC are complemented by reductions in performance elsewhere.

**Figure 1e** shows how additional training data improves the encoding performance of Whisper Large (636 million parameters, layer 30). As before, we fit separate encoding models on increasing amounts of training data. Additional training data for Whisper has an effect that is comparable to OPT: Encoding performance is linearly related to log-dataset size ($r = 0.988$), and increasing the training dataset by an order of magnitude increases performance by 144%.

**Figure 1f** shows the performance of each layer of every Whisper and WavLM model. For legibility, HuBERT results are omitted from this plot and are included in the supplement (Figure B.2). The upper-middle and uppermost layers of each model tend to have the best performance, aligning with previous results on acoustic encoding models [29, 31]. In contrast with WavLM, the Whisper models increase in performance with layer depth; this can likely be attributed to our choice of only using the encoder module from the network.

Voxelwise scaling laws, examining the relationships described in **Figure 1** on a per-voxel basis, can be found in Section A of the supplement.

## 3.3 Large-scale encoding models

After characterizing these scaling laws, we next visualized the performance of one of the top-performing semantic encoding models [1].

**Figure 2** shows the encoding performance of the best OPT model, which uses the 33rd layer of OPT-30B, as measured on the test story with 10 repeats. For several voxels from different areas of cortex we show the encoding model predicted timecourse and ground truth BOLD response. We see strong prediction performance across cortex, with "classical" language regions like Broca's area and auditory cortex being well explained, as well as areas that are typically considered to be more "amodal" in nature, like prefrontal cortex. Voxelwise correlations for this subject are as high as $r = 0.82$. A similar map showing the change in encoding performance from OPT-125M (comparable to GPT models used in earlier papers) to OPT-30B is given in the supplemental material (see Figure C.1).

## 3.4 Noise ceiling analysis

We further investigated the degree to which encoding models can be improved past this point. To do this, we performed a noise ceiling analysis whereby for each voxel, we estimated its $CC_{max}$ (see Section 2.5). This gave us an approximation of the degree to which an ideal encoding model could explain the response variance in each voxel. We then renormalized the correlations from Figure 2 to compute a normalized correlation coefficient $CC_{norm}$.

**Figure 3a** shows the *room for improvement*, or the difference between the correlation coefficients measured in Figure 2 and their $CC_{max}$. Voxels are yellow if there is significant room for improvement, and purple if the model for that voxel is already close to optimal. Regions that are typically believed to contain high-level representations of language such as angular gyrus (AG) [50–52] still have the potential for substantial modeling improvement, while some areas in temporal cortex (near AC), prefrontal cortex (PFC), and the precuneus (PrCu) are nearly optimal. **Figure 3b** shows a histogram of absolute correlation coefficients ($CC_{abs}$), and **Figure 3c** shows the normalized correlations $CC_{norm}$.

---

[1]Keeping with the scaling results from Section 1, we chose to demonstrate this using the best model from the OPT family, however it should be noted that the best model from the LLaMA family is about 5% more performant as measured by correlation. This LLaMA model is further explored in Section 3.6.

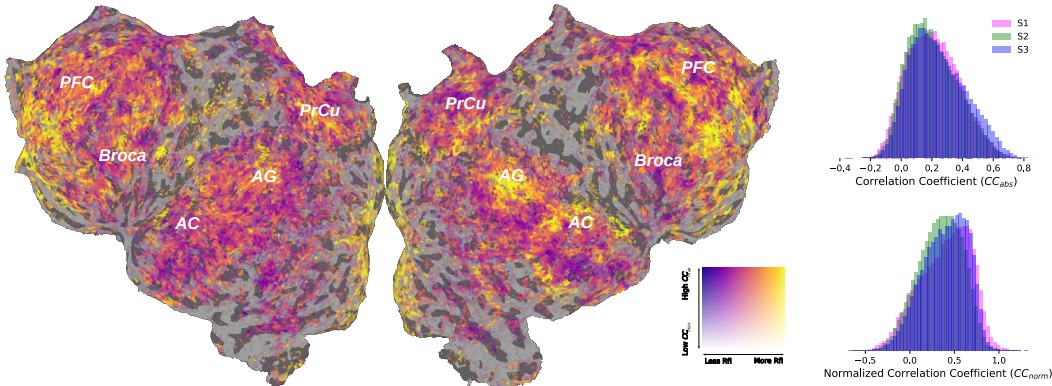

Figure 3: *Noise Ceiling Analysis* - **Figure 3a:** A two channel flatmap showing which ROIs remain poorly explained by an encoding model built from the 33rd layer of OPT30B. Voxels are less transparent if they have a higher idealized encoding performance ($CC_{max}$). Voxels are more yellow if they have high *room for improvement*, defined as the difference between the best possible encoding model and this model. Angular gyrus and some parts of prefrontal cortex are still poorly explained, while precuneus and higher auditory cortex are close to optimal. **Figure 3b:** A histogram of voxel correlations ($CC_{abs}$). **Figure 3c:** A histogram of normalized voxel correlations ($CC_{norm}$). (PFC = prefrontal cortex, PrCu = precuneus, AC = auditory cortex, AG = angular gyrus)

### 3.5 Long context artifacts

Granting encoding models access to contexts as long as 512 tokens implicitly gives them access to the information that the fMRI scan has started recently. For instance, if the input context has only 64 tokens, this implies that the context is occurring at the 64th token in the story. In parallel, responses in some voxels tend to rise or fall gradually over the first minute of each scan (potentially due to underconstrained detrending at scan edges, MRI magnetization reaching steady state, or neural adaptation). The combination of these two effects can have unintended effects on the fair evaluation of these models by artificially inflating measured performance, as encoding models are adept at capturing this early slow drift. We found that long context effects exist up to roughly 100 seconds into a story, so to mitigate this issue we simply exclude the first 100 seconds of predicted and actual responses from each test story when measuring encoding model prediction performance. Figure D.1 in the supplement gives a map of the effect of long-context artifacts on measured encoding model performance. Long-context artifacts can inflate measured performance by up to 20%, but the effects are mostly localized to areas typically associated with low-level speech processing such as early auditory cortex. This effect is most prominent for encoding models using early LM layers and speech models, and tends to not be as significant for later LM layers.

### 3.6 Unifying semantic and speech encoding models with stacked regression

We used stacked regression (see Section 2.4) to augment our best semantic model with the Whisper speech model representations. **Figure 4a** shows the regions that benefit from this augmentation, blended with a flatmap showing the overall semantic encoding model performance. We observe that these benefits are highly localized to auditory cortex and mouth motor cortex. The butterfly plot in **Figure 4b** shows the effect on voxels modified by this augmentation. We see that the auditory cortex voxels that are best predicted by the semantic model are also those that are most improved by this augmentation. **Figure 4c** plots the center of mass of the attribution weights $\alpha^{v,s}$. For voxels where the attribution weights favored the later layers of the Whisper model, the voxel is plotted in a brighter hue. We see that this attribution plot demonstrates a clear progression of auditory information from primary AC to secondary AC coinciding with layer depth. **Figure 4d** shows the benefits of this stacked regression augmentation. We see that the lion's share of the improvements happen in primary AC and early secondary AC.

## 4 Discussion & conclusions

These results suggest the existence of two major effects on the capacity of encoding models to predict BOLD response given finite brain data. First, LM changes that correspond to downstream task

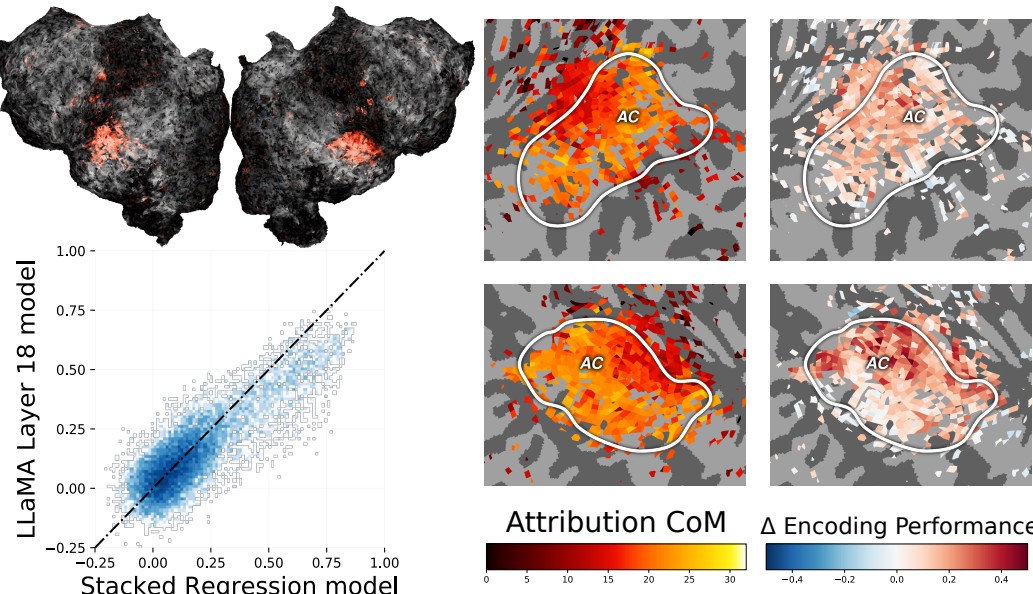

Figure 4: *Stacked Regression* - **Figure 4a:** A flatmap shows which regions of cortex improve when augmenting a semantic encoding model built from the 18th layer of LLaMA-33B with the layers of Whisper using stacked regression. Voxels used the stacked regression if the stacked regression performed better on a validation set. The effect is highly localized to auditory cortex. **Figure 4b:** A butterfly plot comparing the voxelwise encoding performance of the stacked regression encoding model to the baseline semantic model. **Figure 4c:** The center-of-mass of the stacked regression attributions, $\mathcal{C}(\boldsymbol{\alpha}^{v,s})$ are visualized in auditory cortex. **Figure 4d:** The improvement in encoding performance of the stacked regression model over the baseline is visualized in auditory cortex.

performance improvement tend to also improve encoding performance, such as when moving from a LM trained on little data to one trained on more data. Second, increasing hidden state size while keeping other metrics fixed tends to lower encoding performance, as it worsens the conditioning of the encoding model regression problem without a corresponding benefit to model effectiveness. The conflict between these two effects has led to a scenario where the largest model is not necessarily the best for predicting BOLD responses, as we have seen for both the OPT and LLaMA LMs where encoding model performance peaks at about 30B parameters. Rather, a careful balance must be struck between model size and model efficacy in order to maximize encoding performance. Audio models, on the other hand, have not yet seemed to reach this plateau.

What are the use cases for better encoding models? One promising application is the use of encoding models to supplement more classical experimentation techniques, as suggested by Jain et al. [20]. Higher encoding performance leads to more trustworthy model predictions and more accurate conclusions. Another use case of effective encoding models is language decoding, or predicting the language stimulus from the BOLD response. Recent work has shown that effective language decoding models can be built from encoding models by applying Bayesian techniques [45, 53], so it is likely that the performance of such decoders will improve along with the performance of encoding models [33, 44]. Finally, improved encoding performance could enable fine-grained control over voxel activation through stimulus generation, as demonstrated by Tuckute et al. [54].

Given our results, what can computational neuroscientists do to improve the performance of their own encoding models? One potential observation is that *deep* datasets [43, 55–57] — those that focus on collecting many samples from a few subjects, rather than a little data from many subjects — are more useful for modelling brain activity. Encoding performance improvements scale well with both model size and dataset size, and large datasets will no doubt be necessary in producing useful encoding models. Another straightforward adjustment that can be performed is to simply use larger, more performant LMs for building encoding models. To the authors' knowledge, no other natural language encoding models paper at the time of this writing has used models larger than GPT-2 XL, which is a 1.5B parameter model with performance far below the best 30B parameter models. This

could be due to valid concerns that the amount of natural language brain data available is insufficient to train effective encoding models on such a scale. However, we found that even in low data cases, such as with as little as an hour's worth of data, encoding models built from larger models tend to outperform their smaller counterparts, as seen in Figure E.1 of the supplement. We have released code as well as selected precomputed features, model weights, and model predictions generated for this paper. [2] We hope this data release will encourage the use of more performant encoding models in natural language computational neuroscience.

## Acknowledgements

The authors acknowledge and thank the Texas Advanced Computing Center (TACC) at The University of Texas at Austin for providing HPC resources that have significantly contributed to the research results reported within this paper. This work was funded by grants from the NIDCD and NSF (1R01DC020088- 001), the Burroughs-Wellcome Foundation, and a gift from Intel Inc. We thank Ruogu Lin, Leila Wehbe, and Javier Turek for their aid and thoughtful suggestions in assisting with this work.

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
