# A  Voxelwise Scaling Laws

While results presented in the main text of the paper show scaling by averaging across cortex, we can also examine scaling on a per-voxel basis. For a given voxel $v$ we find the line of best fit $\Delta\rho_v \approx m_v \log_2 N$, and then plot $m_v$ which denotes the constant amount by which the correlation at $v$ improves when $N$, the attribute being scaled, doubles. The flatmaps below show parametric and data size scaling across our three subjects.

## A.1  Parametric Scaling

Below are flatmaps showing voxelwise parametric scaling laws, that is, when $N$ is the number of parameters being used in the model used for feature extraction.

### A.1.1  OPT Model Family

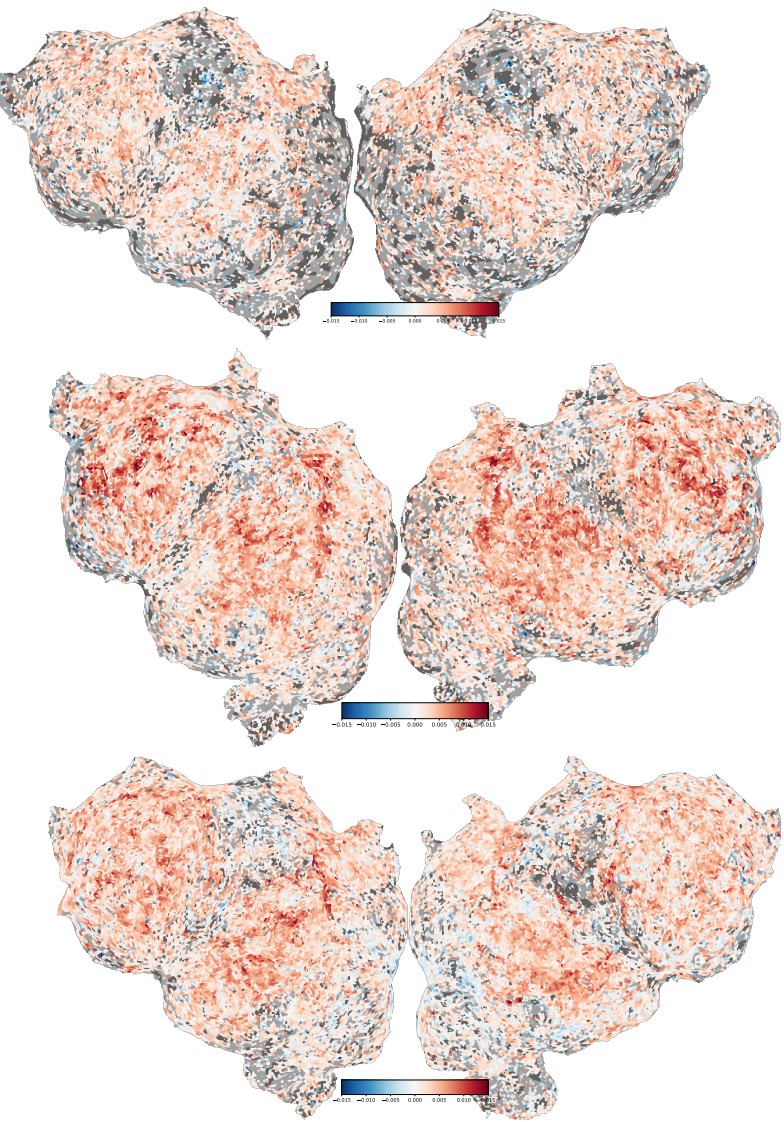

Figure A.1: Parametric voxelwise scaling laws computed using the OPT language model family. Flatmaps show the constant of proportionality of encoding performance for for model size scaling. Model size increases in semantic models seem to be most beneficial for predicting amodal, post-auditory cognitive areas such as prefrontal cortex.

### A.1.2 Whisper Model Family

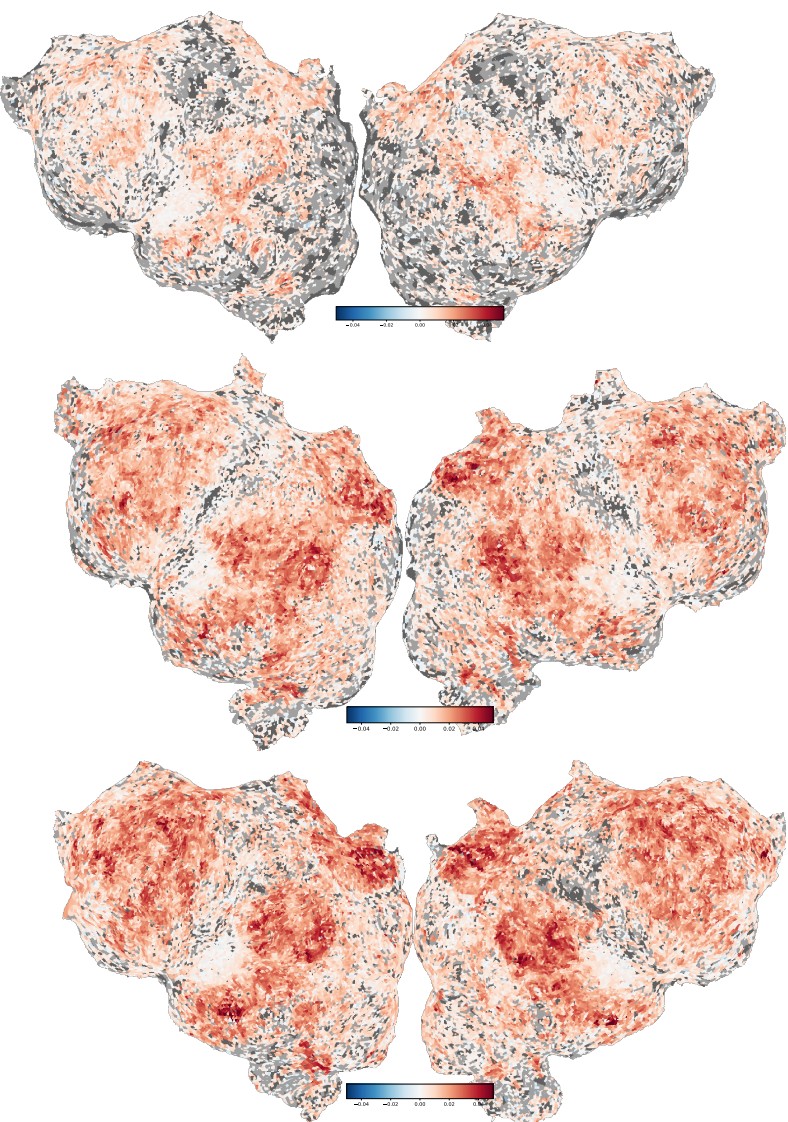

Figure A.2: Parametric voxelwise scaling laws computed using the Whisper audio model family. Flatmaps show the constant of proportionality of encoding performance for for model size scaling. Model size improvements are relatively smaller in auditory cortex, suggesting that the most useful encoded audio features are already captured by the simplest models.

## A.2 Dataset Size Scaling

Below are flatmaps showing voxelwise dataset size scaling laws, that is, when $N$ is the number of stories being used to train the linear weights of the encoding model.

### A.2.1 OPT-30B

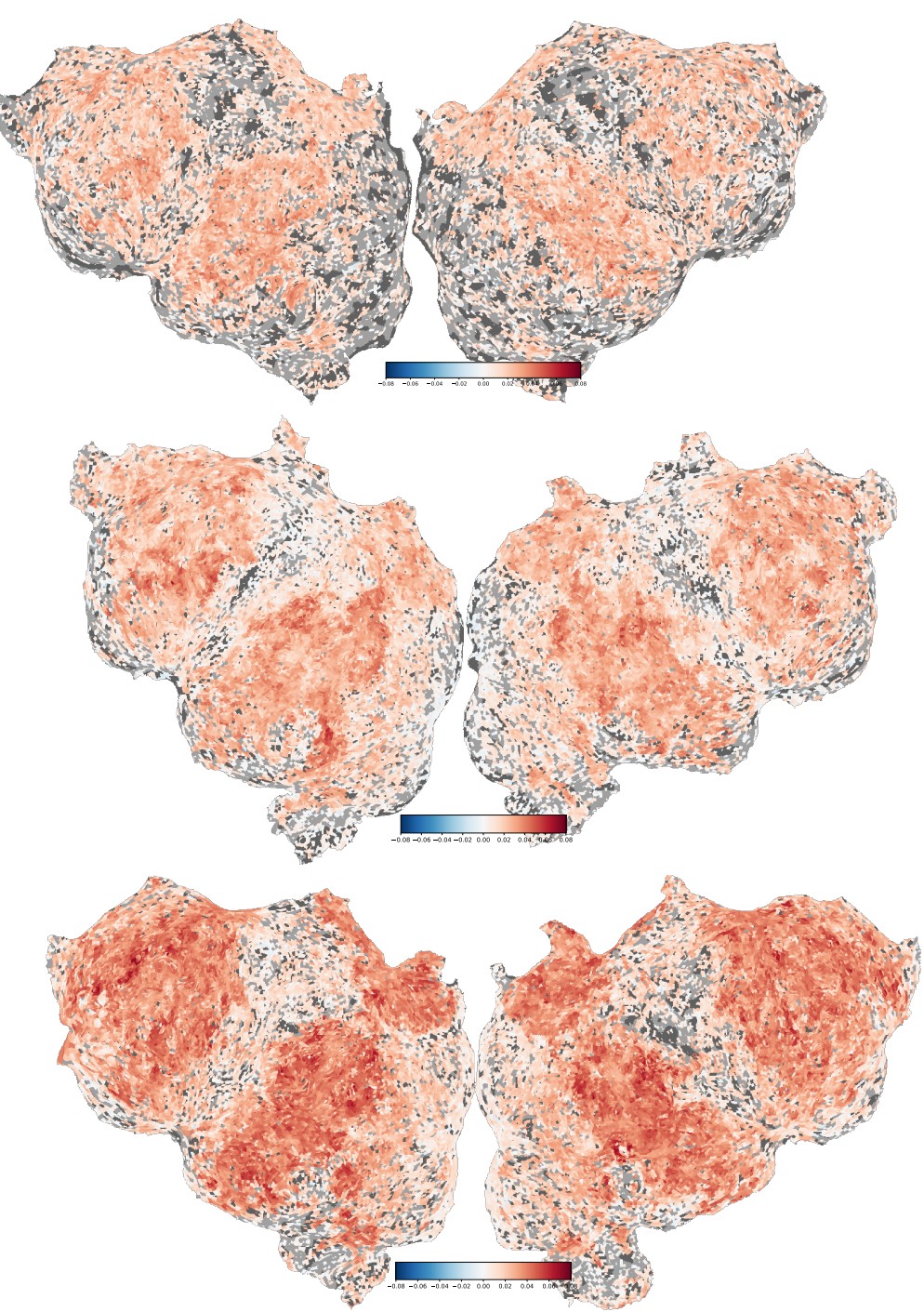

Figure A.3: Dataset size voxelwise scaling laws using OPT-30B. Flatmaps show the constant of proportionality of encoding performance for for dataset size scaling. Dataset size increases in semantic benefit most well-predicted regions without significant spatial preference.

## A.2.2 Whisper-637M

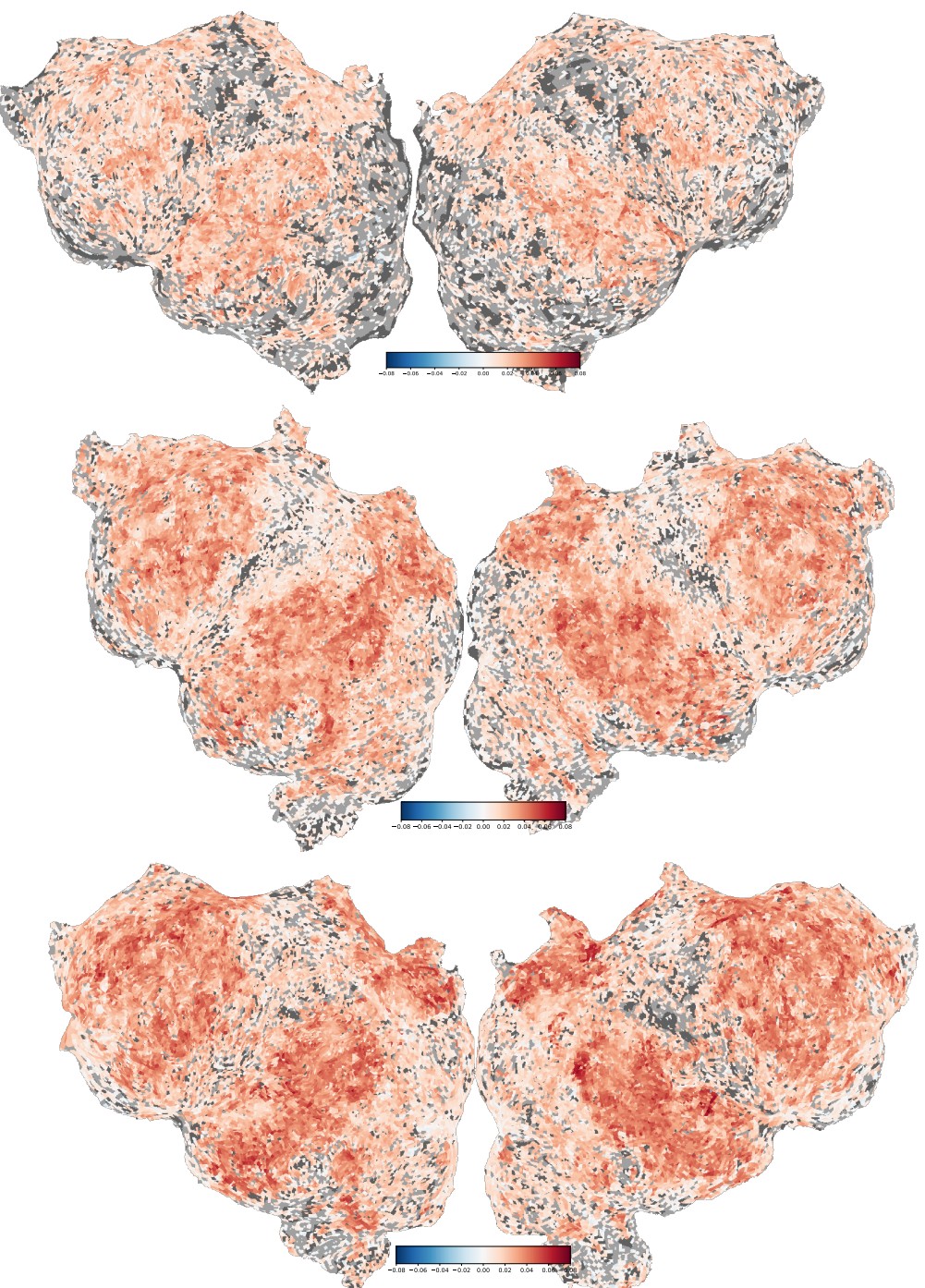

Figure A.4: Dataset size voxelwise scaling laws using Whisper-637M. Flatmaps show the constant of proportionality of encoding performance for for dataset size scaling. Dataset size increases in semantic benefit most well-predicted regions. Certain portions of precuneus and auditory cortex benefit somewhat less from dataset scaling than in OPT.

# B    Scaling laws for speech audio encoding models

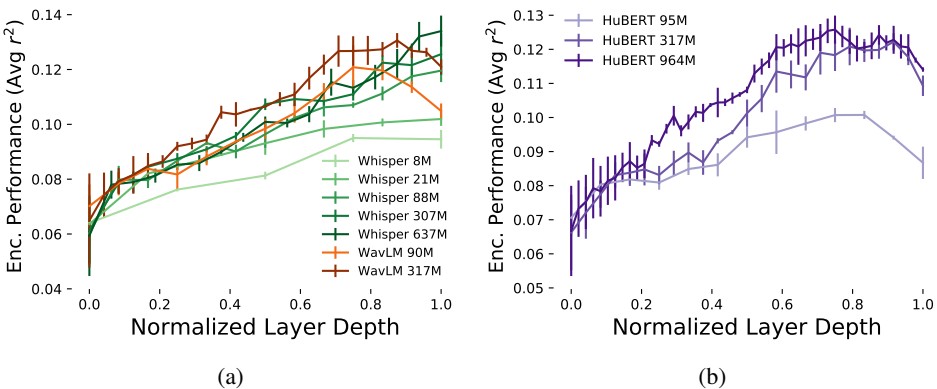

Figure B.1: Performance of audio encoding models, averaged across all voxels in auditory cortex. (a) performance for Whisper and WavLM models. (b) performance for HuBERT models.

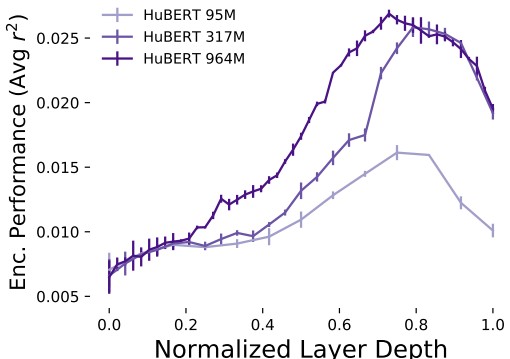

Figure B.2: Performance of HuBERT models, averaged across voxels in cortex. Refer to Figure 1 for Whisper and WavLM models.

## C    Scaling Improvements

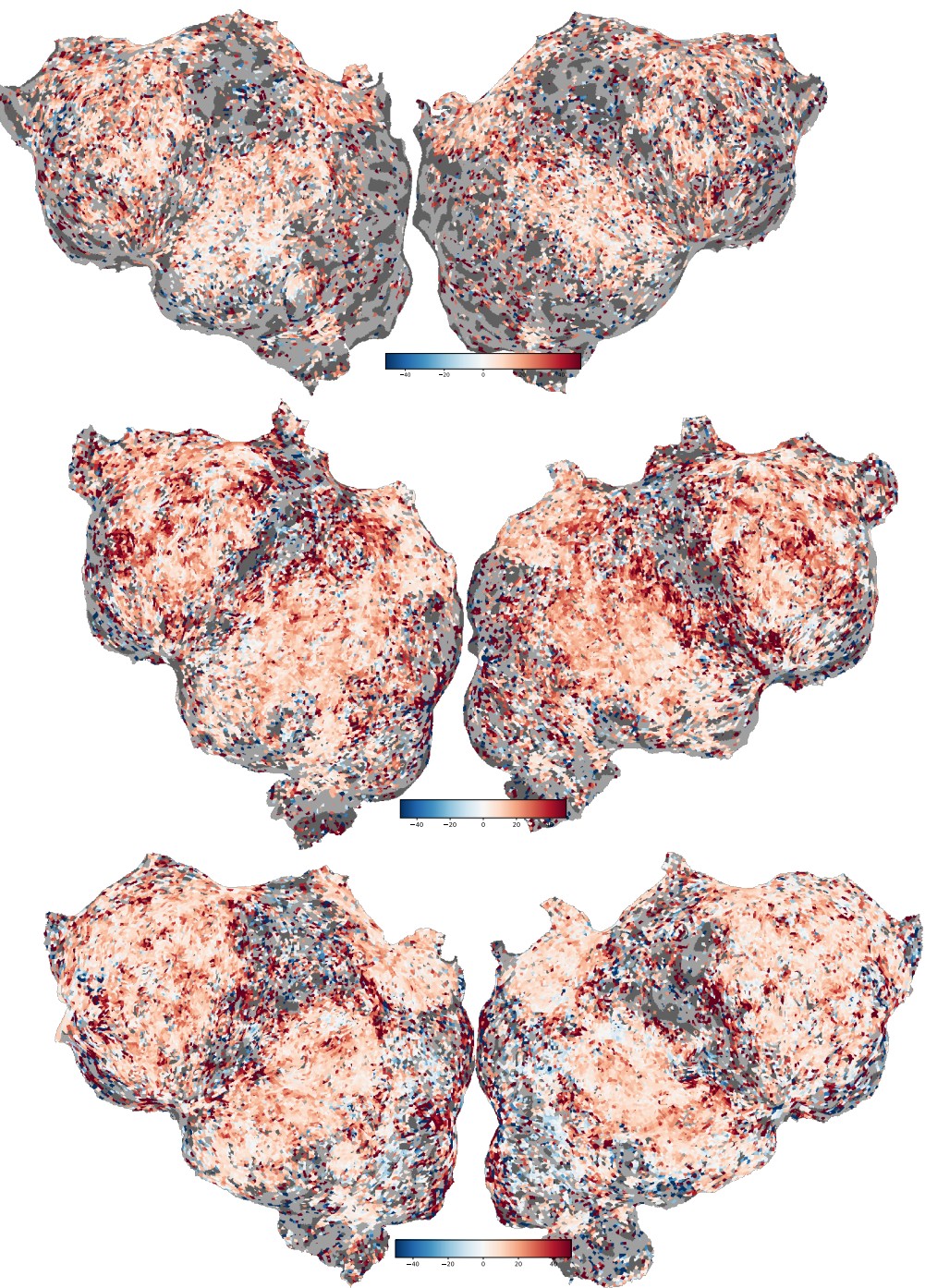

Figure C.1: Percent voxelwise improvements in encoding performance ($CC_{abs}$) from the best OPT-125M layer to the best OPT-30B layer for each of three subjects. We see overall improvement in most areas, with especially large improvements in prefrontal cortex and parietal cortex.

# D    Long context artifact effects

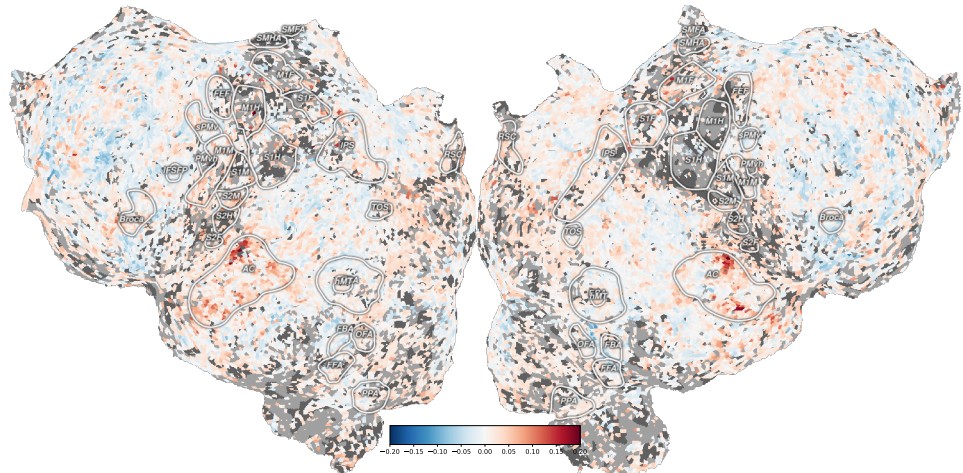

Figure D.1: *Long Context Artifact* - An example of a long context artifact effect as measured on an early layer from OPT-30B (*Uncorrected - Corrected*). The effect is highly localized to primary AC and can lead to bias in encoding performance measurement if not considered.

# E   Joint data-parameter scaling results

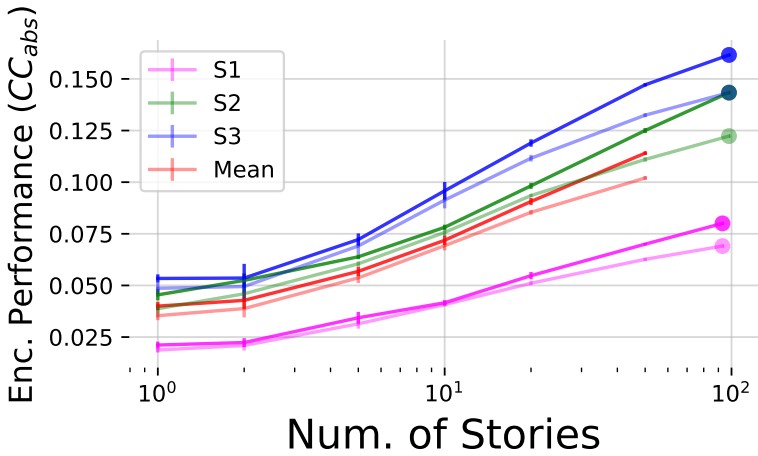

Figure E.1: Comparison of raw encoding performance of the best layers of OPT-125M (*transparent*) and OPT-30B (*bold*) for each of three subjects. We see that OPT-30B consistently outperforms its smaller variant even in the low-data regime of a single story.

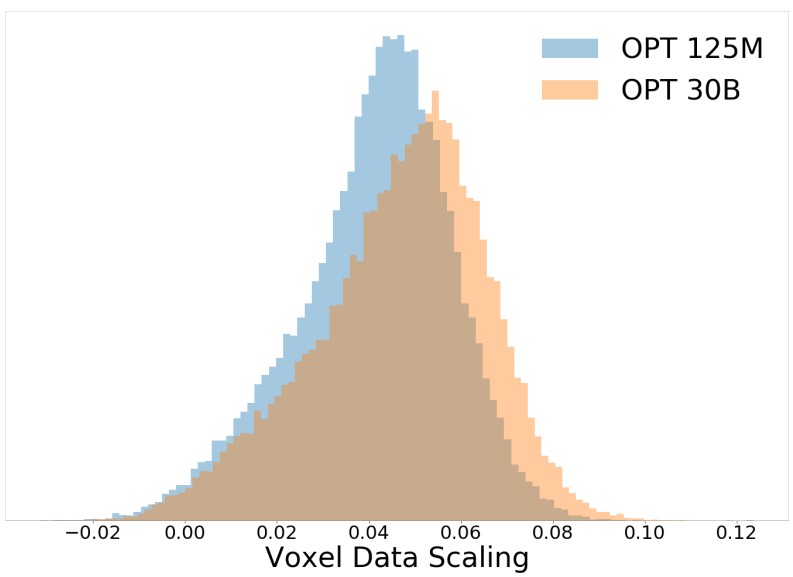

Figure E.2: Histogram showing the slopes of voxelwise scaling laws for two OPT model sizes, shown for S03. As model size increases, the marginal benefit of additional data increases. The relationship between data and parametric scaling suggests a conditioning effect in large-scale encoding models resulting from insufficient amounts of data. Voxels are included if $CC_{max} > 0.5$.

# F Cross-subject results

Flatmaps presented in the main text only used one subject, **S3**. We present analogous flatmaps for the other two subjects, **S1** and **S2**, in this supplemental section.

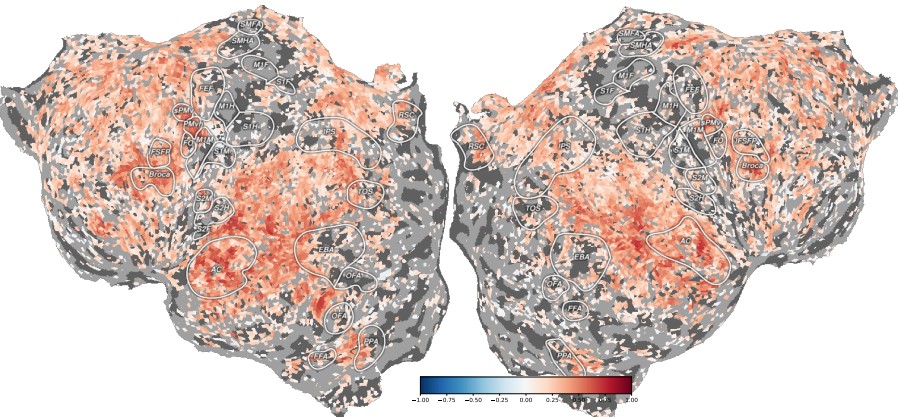

Figure F.1: *Large Scale Encoding Models* - **S1** - Voxelwise correlations using the best OPT-30B layer.

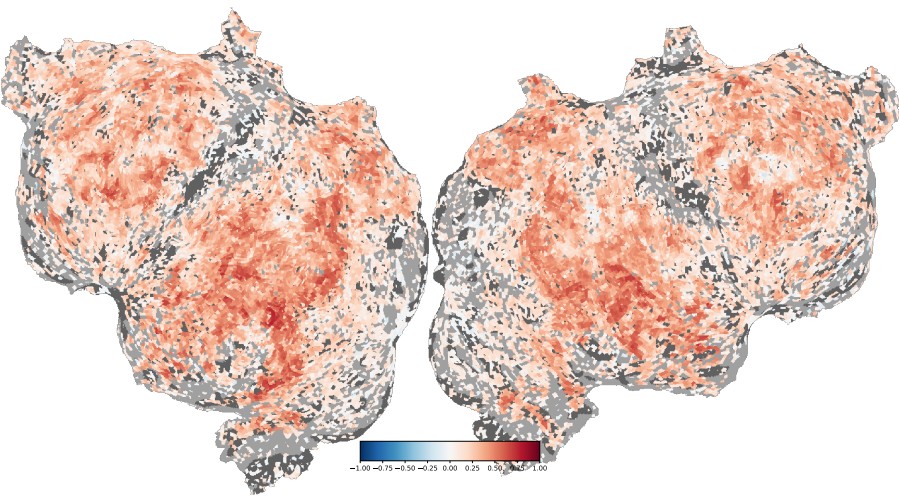

Figure F.2: *Large Scale Encoding Models* - **S2** - Voxelwise correlations using the best OPT-30B layer.

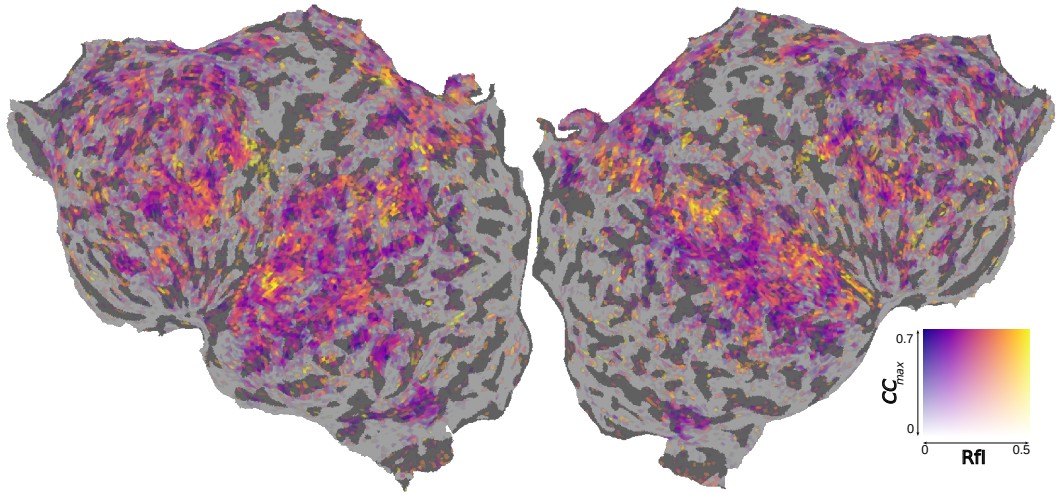

Figure F.3: *Room for Improvement* - **S1** - A two channel flatmap showing which ROIs remain poorly explained by an encoding model built from the best layer of OPT30B.

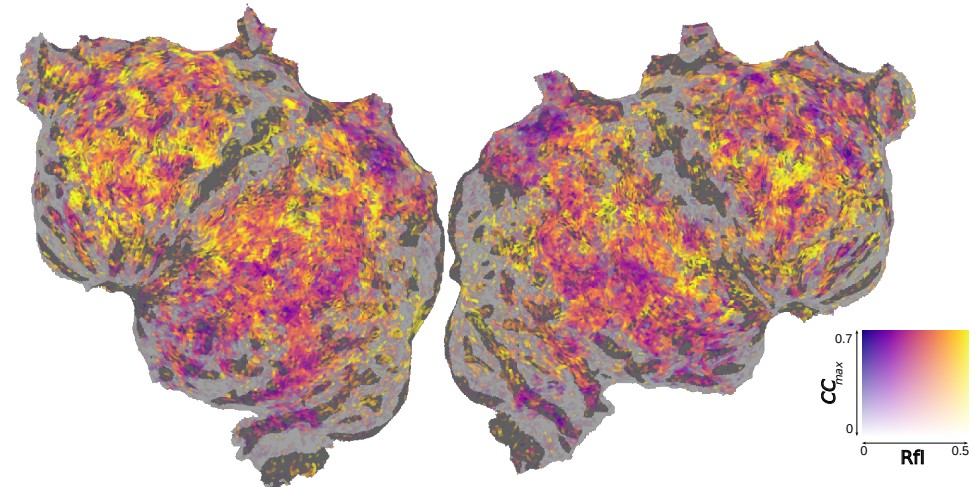

Figure F.4: *Room for Improvement* - **S2** - A two channel flatmap showing which ROIs remain poorly explained by an encoding model built from the best layer of OPT30B.

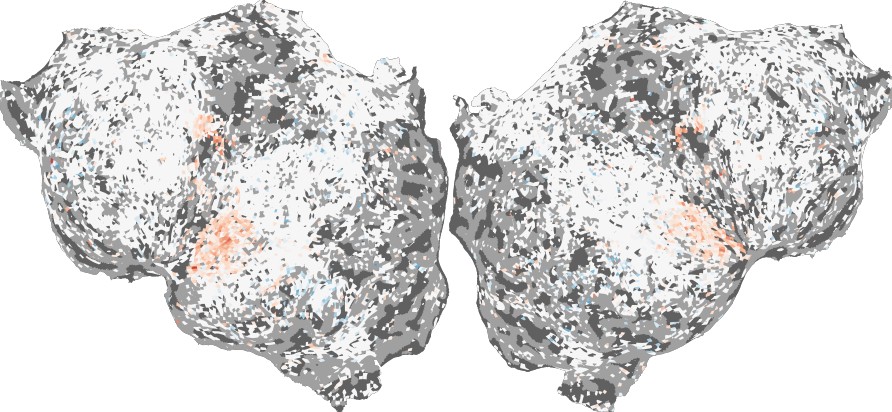

Figure F.5: *Stacked Regression* - **S1** - Improvement over LLaMA baseline using stacked regression with Whisper models

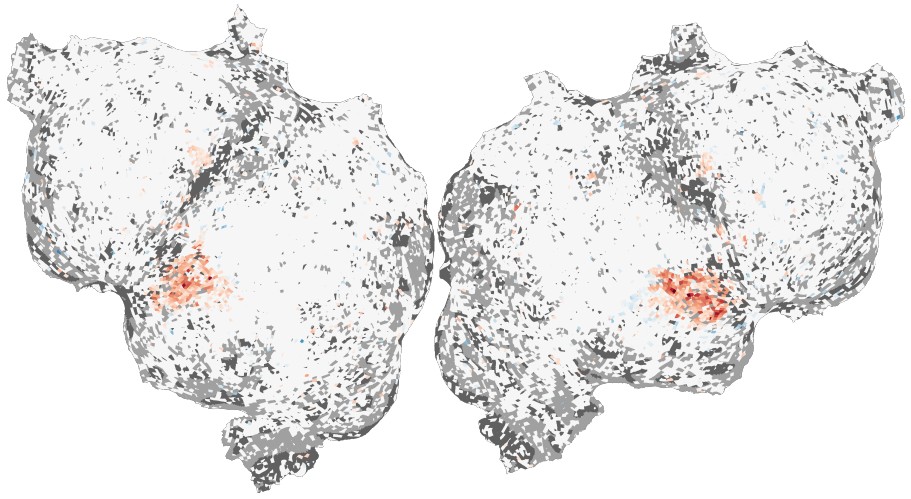

Figure F.6: *Stacked Regression* - **S2** - Improvement over LLaMA baseline using stacked regression with Whisper models

# G  Maximum Correlation Coefficient

Flatmaps of the estimated optimal voxelwise model performance, $CC_{max}$ for each of the three subjects are given below. As these are plots of $CC_{max}$, we do not threhold by $CC_{max}$ as in the other flatmaps.

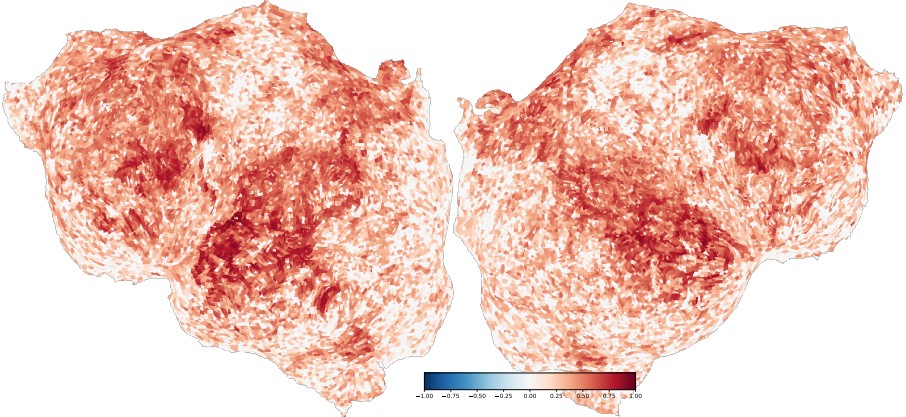

Figure G.1: $CC_{max}$ - **S1**

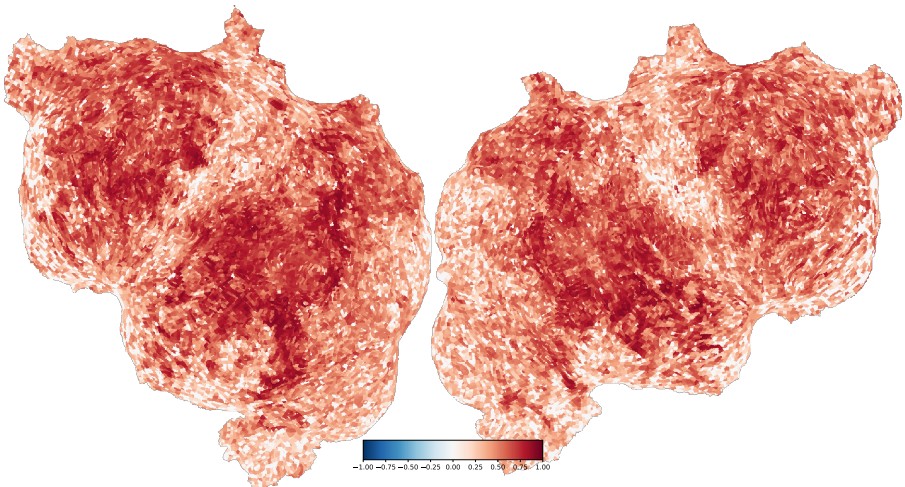

Figure G.2: $CC_{max}$ - **S2**

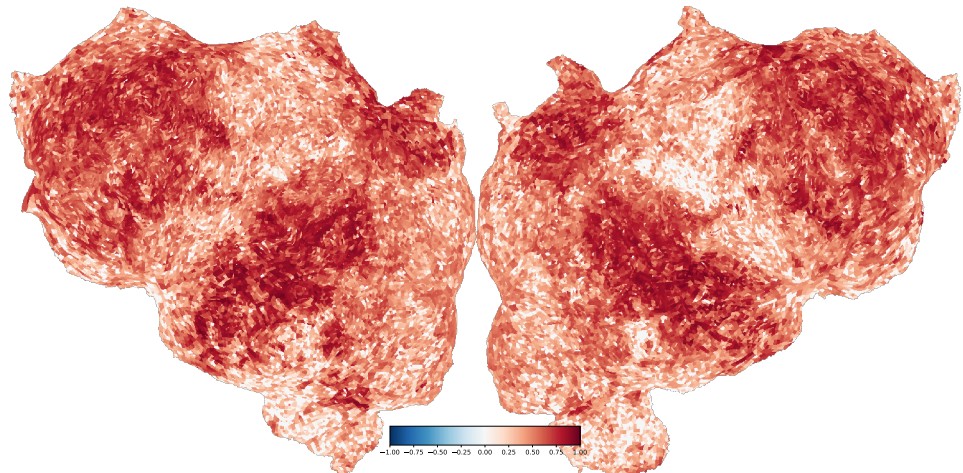

Figure G.3: $CC_{max}$ - **S3**

# H   Stacked Regression Center-of-Mass Attributions

Full flatmaps of the center-of-mass of the attribution weights $\mathcal{C}(\boldsymbol{\alpha}^{v,s})$ are given below (see 2.4).

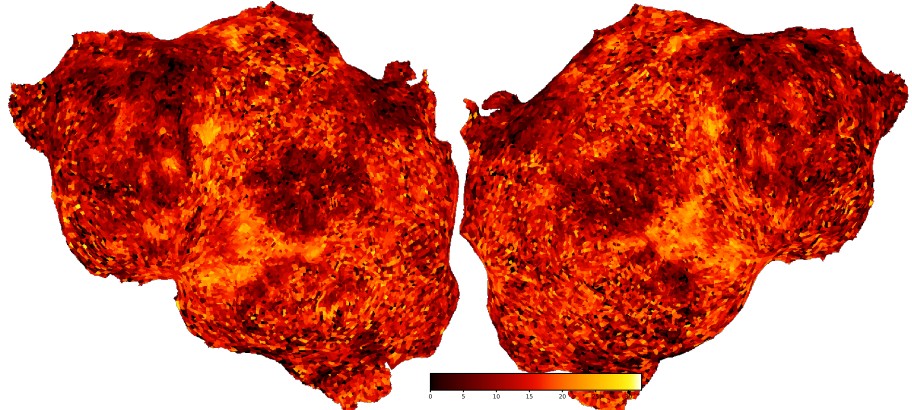

Figure H.1: $\mathcal{C}(\boldsymbol{\alpha}^{v,s})$ - **S1**

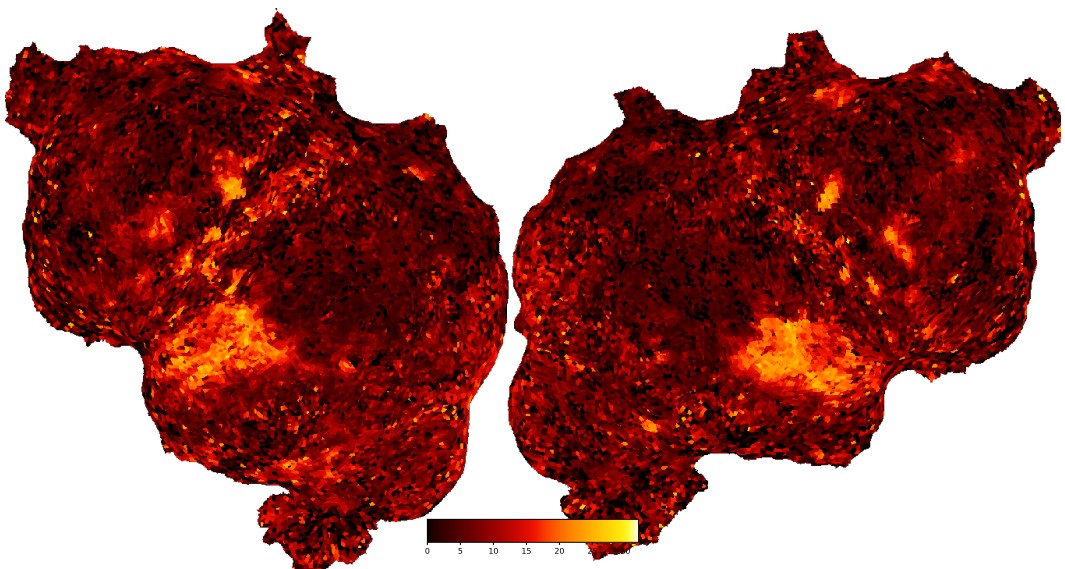

Figure H.2: $\mathcal{C}(\boldsymbol{\alpha}^{v,s})$ - **S2**

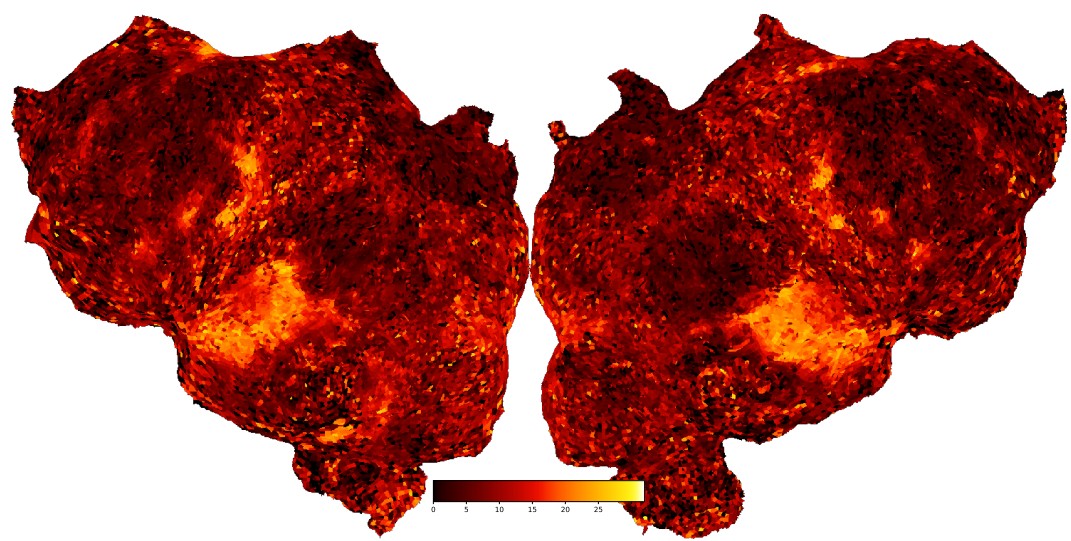

Figure H.3: $\mathcal{C}(\boldsymbol{\alpha}^{v,s})$ - **S3**

# I  Extended Model Details

Table 2: Extended Model Details.

| Family | Layers | Width | Parameters | Perplexity[a] | # Tokens |
|--------|--------|-------|------------|---------------|----------|
| | | | LANGUAGE MODELS | | |
| OPT [38] | 12 | 768 | 125M | 35.91 | 180B |
| | 24 | 2048 | 1.3B | 22.48 | 180B |
| | 40 | 5120 | 13B | 18.17 | 180B |
| | 48 | 7168 | 30B | 17.35 | 180B |
| | 64 | 9216 | 66B | 16.56 | 180B |
| | 96 | 12288 | 175B | DNC | 180B |
| LLaMA [39] | 60 | 6656 | 33B | 10.21 | 1.4T |
| | 80 | 8192 | 66B | 9.73 | 1.4T |

[a]As measured on podcast data. OPT-175B perplexity was not computed due to computational constraints.