# OpenReview forum: "Scaling laws for language encoding models in fMRI"
_NeurIPS.cc/2023/Conference — NeurIPS 2023 poster_

### Official Review · Reviewer_Tsw4 · 2023-07-03

**Soundness:** 3 good
**Presentation:** 4 excellent
**Contribution:** 4 excellent
**Rating:** 7
**Confidence:** 5

**Summary:**

The authors evaluate the effect of LM scale (in terms of N parameters) and fMRI dataset size on the performance of downstream encoding models, trained to predict the activity of individual voxels across the brain as a function of the input those participants received during a scanning session. They explore both the OPT and LLaMa models as pure text based LMs and the HuBERT, WavLM, and Whisper models as acoustic models, across a number of sizes. The authors find that fMRI encoding performance can be improved linearly via logarithmic increases in model scale, up to ~30B for text-based models, and the asymptote has not yet been reached for acoustic models. Increases in the amount of fMRI data included per subject also yield dramatic improvements. They also attempt to characterize what additional encoding performance is yielded by the use of acoustic models over text-based ones and find improvements only in auditory areas but not in higher-level associative areas. Such results propose for increased focus on “deep” neuroscience data collection and scaling up the computational resources utilized for encoding analyses.

**Strengths:**

The work explores a novel and relevant axis for brain encoding analyses and is appropriately contextualized via discussion of related work. The submission is technically sound and most claims are well supported. The submission is clearly written and well organized. The results are relevant and important and will hopefully encourage a “scaling up” of neural encoding analyses.

**Weaknesses:**

The sole evaluation of LM scale as a function of N parameters is reductive and should be addressed. There is a lack or under-specification of uncertainty quantification. See below for more detailed comments.

**Questions:**

General:

Aside from LM model size (N parameters) a crucial component in predicting LM performance is the size of the pretraining dataset (e.g., OPT’s 180B tokens vs. LLaMa’s 1.4T tokens). Given that the authors observe differences between the architecturally similar LLaMa and OPT model families, but these models differ substantially in pretraining dataset size, this is worth mentioning.

Relatedly, for the combined analysis of the LLaMa and OPT models, a more unified axis along which to evaluate the encoding performance of these models might be their mean perplexity across the evaluated podcast stories dataset. I strongly encourage the authors to consider this visualization as a parallel to 1a. It is both likely to be informative, and also proposes a more functional view of the problem space. One would be unlikely to encourage the use of a larger but undertrained 30B model over a smaller 6B model trained on more data with lower perplexity for an fMRI encoding task, particularly due to the author-mentioned advantages of more compressed representations and suggestion that “a careful balance must be struck between model size and model efficacy in order to maximize encoding performance”.

Error bars:

What do the error bars in figure 1b and 1e represent? (stdev, se, x% ci, etc.?)

No uncertainty quantification is included for any other figures. Given that “Encoding model performance for a given layer was computed as the average voxelwise performance of that layer’s hidden states across of all of cortex for all of our 3 subjects”, there would be a number of ways to include uncertainty quantification at this top level even without rerunning any models.

Minor:

increase resolution of figures 1a,1b,1d,1e,3b,3c.


**Limitations:**

The limitations of the work are appropriately addressed.

---

> ### Author Rebuttal · Authors · 2023-08-07
>
> Thank you for your thoughtful review and comments. We have included responses to your questions below.
>
> >Weaknesses:
> >The sole evaluation of LM scale as a function of N parameters is reductive and should be addressed. There is a lack or under-specification of uncertainty quantification. See below for more detailed comments.
> >Questions:
> >General:
> >Aside from LM model size (N parameters) a crucial component in predicting LM performance is the size of the pretraining dataset (e.g., OPT’s 180B tokens vs. LLaMa’s 1.4T tokens). Given that the authors observe differences between the architecturally similar LLaMa and OPT model families, but these models differ substantially in pretraining dataset size, this is worth mentioning.
>
> You are correct, we should have included this detail. We will mention this in an appropriate place in the camera-ready version of this paper.
>
> >Relatedly, for the combined analysis of the LLaMa and OPT models, a more unified axis along which to evaluate the encoding performance of these models might be their mean perplexity across the evaluated podcast stories dataset. I strongly encourage the authors to consider this visualization as a parallel to 1a. It is both likely to be informative, and also proposes a more functional view of the problem space. One would be unlikely to encourage the use of a larger but undertrained 30B model over a smaller 6B model trained on more data with lower perplexity for an fMRI encoding task, particularly due to the author-mentioned advantages of more compressed representations and suggestion that “a careful balance must be struck between model size and model efficacy in order to maximize encoding performance”.
>
> To address your concern, we have computed the mean perplexity across one of our test stories for all but our largest model (which we cannot rerun due to limitations in our computational budget). We would like to note that the perplexities are not directly comparable across the different model families as they employ different tokenization schemes. The table is included below and will be included in some form in the final version.
>
> | Model    | Perplexity |   |   |   |
> |----------|------------|---|---|---|
> | OPT-125M | 35.9191    |   |   |   |
> | OPT-1.3B |  22.4842    |   |   |   |
> | OPT-13B |   18.1750     |   |   |   |
> | OPT-30B |   17.3551     |   |   |   |
> | OPT-66B |   16.5627     |   |   |   |
> | LLaMA-30B | 10.2188     |   |   |   |
> | LLaMA-65B | 9.7363     |   |   |   |
>
> >Error bars:
> >What do the error bars in figure 1b and 1e represent? (stdev, se, x% ci, etc.?)
>
> We apologize for the oversight. These error bars represent standard error. This will be clarified in the camera-ready version.
>
> >No uncertainty quantification is included for any other figures. Given that “Encoding model performance for a given layer was computed as the average voxelwise performance of that layer’s hidden states across of all of cortex for all of our 3 subjects”, there would be a number of ways to include uncertainty quantification at this top level even without rerunning any models.
>
> We chose to omit error bars in Figure 1a and 1d as the only sources of uncertainty quantification would be along the voxel axis or the subject axis. Quantifying uncertainty along the voxel axis would be more misleading than helpful as some voxels are inherently less responsive to language. Quantifying uncertainty along the subject axis is also not necessary here as we plot the individual performance for each subject as well as the mean. Figure 2 similarly shows uncertainty estimation by plotting all individual time courses of the 10 test story repeats.
>
> We will add to the final camera-ready version SNR-normalized subject-axis uncertainty estimation to Figures 1c and 1f, which should give a sense of how the shape of the layerwise encoding performance curve varies from subject to subject. This uncertainty estimation can be seen in Figure 2 of the rebuttal PDF. We are happy to include additional uncertainty estimation in any other figure however we are unsure of where else that information would be beneficial or does not already exist.
>
> >Minor:
> >increase resolution of figures 1a,1b,1d,1e,3b,3c.
>
> Apologies. We intend to revise this for the final camera-ready version.

---

> > ### Comment · Reviewer_Tsw4 · 2023-08-11
> > **Comments addressed**
> >
> > Thanks for addressing this review and for supplying new content.
> >
> > > You are correct, we should have included this detail. We will mention this in an appropriate place in the camera-ready version of this paper.
> >
> > > We apologize for the oversight. These error bars represent standard error. This will be clarified in the camera-ready version.
> >
> > > Apologies. We intend to revise this for the final camera-ready version.
> >
> > Great.
> >
> > > To address your concern, we have computed the mean perplexity across one of our test stories for all but our largest model (which we cannot rerun due to limitations in our computational budget). We would like to note that the perplexities are not directly comparable across the different model families as they employ different tokenization schemes. The table is included below and will be included in some form in the final version.
> >
> > Ah, yes, you are right about tokenization, but thanks for producing this table. Even with that caveat, this is still an informative supplement.
> >
> > > We chose to omit error bars in Figure 1a and 1d as the only sources of uncertainty quantification would be along the voxel axis or the subject axis. Quantifying uncertainty along the voxel axis would be more misleading than helpful as some voxels are inherently less responsive to language. Quantifying uncertainty along the subject axis is also not necessary here as we plot the individual performance for each subject as well as the mean. Figure 2 similarly shows uncertainty estimation by plotting all individual time courses of the 10 test story repeats. We will add to the final camera-ready version SNR-normalized subject-axis uncertainty estimation to Figures 1c and 1f, which should give a sense of how the shape of the layerwise encoding performance curve varies from subject to subject. This uncertainty estimation can be seen in Figure 2 of the rebuttal PDF.
> >
> > Okay, all these decisions are reasonable and the proposed actions look good. Thanks!

---

### Official Review · Reviewer_NDv4 · 2023-07-04

**Soundness:** 3 good
**Presentation:** 4 excellent
**Contribution:** 3 good
**Rating:** 6
**Confidence:** 4

**Summary:**

Researchers compared the effectiveness of larger open-source language models, such as those from the OPT and LLaMA families, in predicting brain responses recorded using fMRI. They found that brain prediction performance improves logarithmically with model size, with a 15% increase in encoding performance as model size increases. Similar improvements were observed when scaling the size of the fMRI training set. The study also explored the scaling of acoustic encoding models and found comparable improvements. The analysis suggests that increasing the scale of both models and data will lead to highly effective models of language processing in the brain, enabling better scientific understanding and applications such as decoding.

**Strengths:**

- Presents a novel empirical observation of scaling laws for language and audio encoding models in fMRI.
- They show log linear scaling in brain prediction, encoding performances in both language and acoustic domains.
- They also show when scaling the size of the fMRI data, the performances increase log-linearly.

**Weaknesses:**

Most of my concerns are addressed in the questions section.
One concern I have is that the code is not publicly available. If the authors published the code it would be greatly appreciated.

**Questions:**

- Wernicke's area is often paired and observed alongside Broca's area. Did the authors consider observing the Wernicke's area for both auditory and language encoding models?
- In section 3.4 figure 3a, the "room for improvement" seems to be slightly more prominent in the right hemisphere. Please note that this observation is purely from the figure. Have the authors explored why this may be the case?
- To observe the scaling laws in increasing data size, the authors withheld different amounts of the training data. Did the authors consider augmenting the data in other ways?

**Limitations:**

The authors adequately addressed the limitations of the paper.

---

> ### Author Rebuttal · Authors · 2023-08-07
>
> Thank you for your review and thoughtful comments. We have included responses to your questions below.
>
> >Weaknesses:
> >Most of my concerns are addressed in the questions section. One concern I have is that the code is not publicly available. If the authors published the code it would be greatly appreciated.
>
> While not currently available, we intend to include a link to code as well as pretrained encoding models that can be used to help replicate our results in the camera-ready version.
>
> >Questions:
> >Wernicke's area is often paired and observed alongside Broca's area. Did the authors consider observing the Wernicke's area for both auditory and language encoding models?
>
> The area we have defined as “left auditory cortex” is also referred to as Wernicke’s area. We apologize for the confusion and will mention this in an appropriate place in the camera-ready version.
>
> >In section 3.4 figure 3a, the "room for improvement" seems to be slightly more prominent in the right hemisphere. Please note that this observation is purely from the figure. Have the authors explored why this may be the case?
>
> This seems to be a subject-specific effect, as the other two subjects (presented in Appendix E) do not have the same lateralization. Given the small number of subjects, it is difficult to productively speculate on the reason for this.
>
> >To observe the scaling laws in increasing data size, the authors withheld different amounts of the training data. Did the authors consider augmenting the data in other ways?
>
> There may be other ways of withholding data, such as based on its semantic content or original source, to analyze the effects on model performance, if that is what you mean by augmentation. However, we think such analyses would likely be beyond the intended scope of this work.

---

> > ### Comment · Reviewer_NDv4 · 2023-08-20
> >
> > Thank you for the clarification! I will keep my score. I wish the authors the best of luck!

---

### Official Review · Reviewer_fnMd · 2023-07-05

**Soundness:** 3 good
**Presentation:** 3 good
**Contribution:** 2 fair
**Rating:** 7
**Confidence:** 5

**Summary:**

The authors study how the scaling of large language models produce accurate features to predict human brain fMRI activity. They report scaling-like laws for fMRI encoding models – models with more parameters tend to more accurately predict fMRI activity (as measured with the Pearson correlation of the encoding model predictions and fMRI time series).

**Strengths:**

I found the paper to be of general interest, and the methods to be sound. The paper suggests that there is a utility to using modern large ML models as producing features for fMRI encoding models.

**Weaknesses:**

It is hard to determine whether this result should be expected, and what to make of its scientific significance (beyond an engineering feat). On one hand, one would expect that models that are larger (and therefore better at modeling language/sound data) should produce more useful features for language/sound data. On the other hand, these large models process language/sound data in ways that are almost certainly distinct from how the brain processes information (e.g., the brain doesn’t appear to merely do next-word prediction). So what does one make of this discrepancy? Do the authors believe that the brain mimics the computational processing of these models? And what is the scientific significance of using a model – that presumably looks nothing like the brain – to build better models of the brain?

How valuable is using percent correlation change as a metric for how well the encoding models perform relative to, for example, R^2? Is it truly useful to benchmark or establish ‘scaling laws’ based on the average encoding performance across all fMRI voxels, rather than identify one or two key regions (or the variability across regions)? What might be interesting to evaluate is whether these scaling laws only apply to a subset of voxels/brain regions, since, presumably, how well a voxel/region follows a scaling law is likely dependent on how well a brain region’s function matches with the model type.


**Questions:**

For an ‘fMRI encoding model practitioner’, it would be helpful if there were concrete ‘recommendations’ for what models are most appropriate for language/audio modeling. For example – and this is a minor point – does it matter which layers one uses to produce the regression model features?

If possible, it would be interesting to see a surface map that shows which voxels/regions follow a scaling law for the audio v language models, respectively.

How dependent is the scaling law on computing the average (across voxels) for each of the maps per model?

Can the authors also report R^2 for their encoding models benchmark?

What does it mean (scientifically) that the model that best produces regression features for brain encoding models look nothing like the brain? Are there conclusions one should draw from this? Or is this just a coincidence?

Might there be any correspondence between layers in an LLM and different brain regions?

**Limitations:**

The limitations are mostly in terms of the scientific utility of claiming that there are scaling laws for fMRI. I doubt the authors intend this interpretation, but the title seems to hint at the possibility that if one achieves enough scale with language models, one might be able to perfectly predict brain activity (and therefore model the brain). This seems implausible (and likely not intended), but some discussion around this would be helpful.

---

> ### Author Rebuttal · Authors · 2023-08-07
>
> Thank you for your review and thoughtful comments. We have included responses to your questions below.
>
> >Weaknesses:
> >It is hard to determine whether this result should be expected, and what to make of its scientific significance ... So what does one make of this discrepancy? Do the authors believe that the brain mimics the computational processing of these models? And what is the scientific significance of using a model – that presumably looks nothing like the brain – to build better models of the brain?
>
> We thank the reviewer for raising this point, as it is an important one. The suggestion that language models share fundamental commonalities with human brains is an ongoing discussion - see Schrimpf et. al., Goldstein et. al., and Caucheteux et. al., c.f. Antonello and Huth for more details. However, regardless of whether these models are mechanistically similar to the brain, they remain the most powerful fine-grained predictive models of biological language processing that have ever existed, and that alone gives them high scientific and practical value. While prediction should not be conflated with explanation, powerful predictive models of brain activity provide a uniquely useful pathway to eventual explanation. Sufficiently powerful encoding models can be used to simulate brain activity *in silico*, allowing for the rapid simultaneous testing of thousands of hypotheses that would be impossible to test *in vivo*. Furthermore, encoding models play an important role in Bayesian decoding techniques which have tremendous practical value – see Tang et al. 2023 for further discussion.
>
> > How valuable is using percent correlation change as a metric for how well the encoding models perform relative to, for example, R^2? Is it truly useful to benchmark or establish ‘scaling laws’ based on the average encoding performance across all fMRI voxels, rather than identify one or two key regions (or the variability across regions)? What might be interesting to evaluate is whether these scaling laws only apply to a subset of voxels/brain regions, since, presumably, how well a voxel/region follows a scaling law is likely dependent on how well a brain region’s function matches with the model type.
>
> Appendix B in the supplement gives a voxelwise breakdown of parameter scaling as a percentage improvement from our OPT-125M model to our OPT-30B model. As an additional analysis, we have now also generated flatmaps of the slopes of the individual voxelwise scaling laws for both data scaling and parameter scaling (see Figures 3 and 4 in the rebuttal PDF). We see that for data scaling, almost all voxels associated with language processing in cortex benefit from additional data. Scaling laws for parameters are a bit more complicated, but it seems that in general, areas that are typically associated with high-level language processing and cognition are those that benefit the most from increasing parameter size.
>
> >For an ‘fMRI encoding model practitioner’, it would be helpful if there were concrete ‘recommendations’ for what models are most appropriate for language/audio modeling. For example – and this is a minor point – does it matter which layers one uses to produce the regression model features?
>
> We believe that this is addressed in the current draft – see Figures 1c and 1f which show which layers perform best in each model. We would recommend layer 18 from the LLaMA-30B model and layer 20 from WavLM-317M, as these layers perform well and can be computed with relative ease. In cases with larger amounts of data than we possess, we would suggest using representations from larger models. We will add an explicit mention of these suggestions in the final version. Pretrained weights of these models will be released as well in the camera-ready version.
>
> >If possible, it would be interesting to see a surface map that shows which voxels/regions follow a scaling law for the audio v language models, respectively.
>
> Yes, we will include this in the supplement, and it has been attached to this rebuttal.
>
> >How dependent is the scaling law on computing the average (across voxels) for each of the maps per model?
>
> Some voxels scale more than others, but scaling is a brain-wide phenomenon that impacts nearly all language-selective brain regions positively. This is observable from the flatmaps mentioned above.
>
> >Can the authors also report $R^2$ for their encoding models benchmark?
>
> Unfortunately, these models were trained to optimize correlation and not $R^2$, so computing $R^2$ directly would not be meaningful. Retraining the models with the $R^2$ objective would fall outside our compute budget, but we will include a plot similar to Figure 1c showing the average $r^2$ across voxels, which bounds $R^2$ from above. This can be found in the rebuttal PDF (Figure 2). We will also be correcting a notational oversight and converting occasional mentions of $R$ to $r$.
>
> >What does it mean (scientifically) that the model that best produces regression features for brain encoding models look nothing like the brain? Are there conclusions one should draw from this? Or is this just a coincidence?
>
> We would not describe it as a coincidence. It makes sense that two systems that can both engage with language in a high-level way will have shared information about that language. This does not entail that they are performing the same computations or are “learning” in the same way. Your question is part of a larger scientific debate about encoding models that extends beyond the scope of this particular paper.
>
> >Might there be any correspondence between layers in an LLM and different brain regions?
>
> For language models we don’t find this to be the case. However, for audio models there does seem to be some correspondence, where earlier layers predict primary auditory cortex better and later layers predict higher-order auditory cortex better. This is shown in Figure 4c, with the center-of-mass attribution analysis in the stacked model.

---

> > ### Comment · Reviewer_fnMd · 2023-08-14
> >
> > I thank the authors for their response. I think the results of this paper are of useful scientific fact and value to the greater community. I raise my rating from 6 to 7.

---

### Official Review · Reviewer_jqqd · 2023-07-06

**Soundness:** 4 excellent
**Presentation:** 3 good
**Contribution:** 4 excellent
**Rating:** 8
**Confidence:** 3

**Summary:**

Previous works have demonstrated that activations from Language Models fit, to some extent, brain activations in participants listening to audio texts.  Here, the authors examine how the fit is influenced by model size (number of parameters) and the training dataset size.
The main observation is that performance scales log-linearly with model size (until 30B parameters) and with fMRI training set size (from 1 to 100 stories).



**Strengths:**

The main originality of the work is to explore language models that are much bigger than the ones that had been used in previous works of this type (as well as recent, large, acoustic models). Not only does this represent a technical feat, but it also advances knowledge on the topic, as it was not clear if the representations constructed by larger models would necessarily converge towards more and more brain like representations when the models become bigger.  Data presented here show that there was still a lot of room for improvment from GPT-2 like models.

The raw performance of the best model (OPT-30B), presented in Fig.2 is quite impressive and, along with the analysis of the effect offmri training set  size, buttresses the authors' claim of the usefulness of using large within subject dataset.

Fig.3b which compares performance to estimated ceilings is very interesting theoretically as it suggest that the models model some regions better than others.





**Weaknesses:**

As the authors mention (at line 300), when models grow in size, the regression problem may become ill conditioned. Would there be any strategy to remedy this problem?

**Questions:**

Figure 1 shows the increase in performance averaged _across the whole cortex_. Have you checked whether this profile is similar in different regions (e.g. auditory cortex, core language region, and higher-level regions like precuneus and medial prefrontal)?

Were the (OPT) models of different sizes trained with the same language corpora (fixed size)?



**Limitations:**

The limitations were adequately discussed.

---

> ### Author Rebuttal · Authors · 2023-08-07
>
> Thank you for your review and thoughtful comments. We have included responses to your questions below.
>
> > Weaknesses:
> >
> >As the authors mention (at line 300), when models grow in size, the regression problem may become ill conditioned. Would there be any strategy to remedy this problem?
>
> Naturally, poor conditioning is difficult to remedy algorithmically without relying on additional data as nothing will get around the fact that there is insufficient data to mathematically constrain the regression problem. Still, there may be alternative methods for enabling the continued scalability of encoding models without relying on unreasonably sized single-subject datasets. For example, instead of using ridge regression for the linear readout, one might consider using a bottlenecked neural network trained simultaneously on data from multiple subjects to assist in dimensionality reduction. This may help with regularization as it enables the model to learn a general, low-dimensional joint semantic representation that works well across many subjects. We believe that the possibility of jointly training language encoding models across subjects is worthy of additional study.
>
> >Questions:
> >
> >Figure 1 shows the increase in performance averaged across the whole cortex. Have you checked whether this profile is similar in different regions (e.g. auditory cortex, core language region, and higher-level regions like precuneus and medial prefrontal)?
>
> We have, and appendix B gives some indication of the percentage scaling improvements for each voxel/ROI. We agree that this may not be fully addressed, however, and will include an additional flatmap showing the slope of the scaling law per voxel in the supplement (Figures 3 and 4 in the rebuttal).
>
> >Were the (OPT) models of different sizes trained with the same language corpora (fixed size)?
>
> Yes, according to the original OPT paper, all models were trained with the same training set of 180B tokens. We will include a clarification of this detail in an appropriate place in the main text.

---

> > ### Comment · Reviewer_jqqd · 2023-08-18
> >
> > Thanks for the clarifications: I confirm my original rating (strong accept)

---

### Official Review · Reviewer_dAos · 2023-07-10

**Soundness:** 3 good
**Presentation:** 3 good
**Contribution:** 3 good
**Rating:** 5
**Confidence:** 5

**Summary:**

The authors of this paper delve into the investigation of the scaling law between the performance of predicting brain activity (measured using BOLD) and the number of parameters in large language models, coupled with the amount of training data for the linear readout for retrained LLM used in AI. This pursuit of exploring a log-linear scaling law relationship is an interesting and valuable. It holds promise in providing crucial information for paving a path forward in developing better functional models of brain activity, which could be instrumental in understanding brain computations. The technical dimensions of the paper are well-executed. The authors have displayed a solid grasp of the subject matter and have employed appropriate methodologies to carry out their research. However, I have some reservations about the validity of their claims concerning the non-saturation of the scaling law for the number of parameters in large models. Specifically, Figure 1, which is central to their argument, seems to present evidence contrary to their claim. They acknowledge a visible saturation but attribute this observation to memory limitations in the way they fit the linear readout (i.e. they select one layer). However, unless they successfully demonstrate that the saturation does not occur when a more sophisticated linear readout is utilized (for example, from all layers for larger models), it is not justified for them to assert that the predictive models demonstrate scaling as a function of parameter size. Thus, the authors need to refine their claims or provide additional analysis to convincingly demonstrate that the saturation effect they observed is truly an artifact of their methodological limitations, not an inherent feature of the models they are studying. Without such demonstrations, the claim about the non-saturation of the scaling law remains unsubstantiated. In conclusion, while the paper tackles an intriguing concept and showcases strong technical elements, the authors need to strengthen their assertions through concrete evidence or refinement of their conclusions. This additional effort will greatly enhance the paper's overall quality and the validity of their findings, adding substantial value to the body of knowledge in the field.

**Strengths:**

The question of how far scaling deep learning models can continue to improve neural prediction is an important question in the quest to build more accurate models of the brain.

**Weaknesses:**

More work is needed to demosncate that the current LM are not saturating the performance of the neural prediction task. This is very critical since the main contribution of this work is that even the largest LM do not saturate the performance.

**Questions:**

The authors need to demonstrate using other linear readouts that there is not saturation of performance in the neural prediction task.

**Limitations:**

yes

---

> ### Author Rebuttal · Authors · 2023-08-07
>
> >More work is needed to demonstrate that the current LM are not saturating the performance of the neural prediction task. This is very critical since the main contribution of this work is that even the largest LM do not saturate the performance.
>
> Thank you for your review and thoughtful comments. We agree that we did not adequately prove that the performance plateau after 30B parameters is a side-effect of the poor conditioning of the regression problem at that size, so we have performed an additional analysis to strengthen our argument. Doing a more sophisticated linear readout from multiple layers is currently out of reach, as this would be too costly to realistically train and would have even worse conditioning, with the number of features far exceeding the number of training points from our fMRI dataset. Instead, we performed an additional analysis regarding the relationship between data scaling and parameter scaling. Conditioning of the encoding regression problem depends on both the size of the dataset and the number of parameters: more data is good, more parameters is bad. If the problem with the large models is not poor conditioning, then we would expect dataset size to be less impactful than for smaller models. Conversely, if conditioning is the culprit, then we would expect the large models to show stronger data size scaling than the smaller models. Our analysis shows the latter: for larger language models performance scales more strongly with dataset size. Figure 1 of the rebuttal PDF shows a histogram with the distribution of voxelwise scaling laws (i.e. the slope of the log-linear relationship) across cortex for one subject, filtered by a minimum $cc_{max}$ (i.e. only in voxels with sufficient signal to model).
>
> We see that the distribution is shifted to the right for the larger language model, showing that encoding models built from larger LMs on average benefit more from additional data. This makes sense, as large models are those that are most affected by poor conditioning.  As it seems unlikely that the representations from larger language models contain less information than those of smaller models, we suggest that the plateau we observe at 30B is mostly the result of limited dataset size. Given this result, we intend to refine and clarify the paper text, suggesting that it has to do mostly with data constraints and that with sufficient amounts of data, parameter scaling would likely continue to improve above 30B parameters.
>
> We found this result by comparing data scaling between the 125M and 30B OPT models, which we had already fit. Ideally we would have also tested the 175B OPT model, but testing dataset scaling is computationally expensive and re-fitting those models is substantially beyond our compute budget.
>
> Furthermore, in responding to another reviewer, we recomputed Fig. 1c with a sum of $r^2$ metric (Figure 2 in the rebuttal PDF) and found that by using this metric, the model trained by OPT-175B is slightly better than the OPT-30B metric. The sum of $r^2$ metric favors increases in encoding performance in well-predicted voxels over poorly predicted voxels. As well-predicted voxels will be more resilient to poor conditioning, this result aligns with our conditioning argument as well.
> We hope that this argument is satisfactory to you and allays your concerns.

---

### Author Rebuttal · Authors · 2023-08-07

Thank you to all our reviewers for their detailed and thoughtful feedback. For our general response, we are attaching a series of figures that were requested or that we believe resolves the outstanding concerns that each of you have voiced. The figures will each be included somewhere appropriate in the supplemental material of the final camera-ready version. Here are descriptions of the figures, in order:

Figure 1. A histogram of the relationship between data scaling and parameter scaling, meant to flesh our our argument surrounding the conditioning of our ridge regression past the 30B plateau we observe. We see that as model size increase, the average slope of voxelwise data scaling laws increase as well.

Figure 2. A recreation of Figure 1c using an average $r^2$ metric. We see that using this metric, the performance of the best layer in OPT-30B is slightly better than the performance of the best OPT-175B layer, which further supports our point about conditioning.

Figure 3. A flatmap of voxelwise data scaling laws. The redness of the voxel indicates the degree to which that voxel improves with additional data.

Figure 4. A flatmap of voxelwise parameter scaling laws. The redness of the voxel indicates the degree to which that voxel improves with larger language models.

We have attempted to address specific points mentioned in your reviews in each of the reviewer-specific rebuttals. Thanks again for your feedback.

---

### Decision · Program_Chairs · 2023-09-21

**Decision:**

Accept (poster)

**Comment:**

The 5 reviewers here have expressed support on this work and did endorse it for publication. Given the thorough reviews and later discussions I support the publication of this work at NeurIPS 2023.